# A popcorn-inspired strategy for compounding graphene@NiFe$_2$O$_4$ flexible films for strong electromagnetic interference shielding and absorption

Mingjie Liu[1,5], Zhiyuan Wang[1,5], Zhaoqiang Song [2,3,5], Fangcheng Wang[1,4], Guangyao Zhao[1], Haojie Zhu [1], Zhuofei Jia[1], Zhenbin Guo [2] ✉, Feiyu Kang [1] & Cheng Yang [1] ✉

Compounding functional nanoparticles with highly conductive and porous carbon scaffolds is a basic pathway for engineering many important functional devices. However, enabling uniform spatial distribution of functional particles within a massively conjugated, monolithic and mesoporous structure remains challenging, as the high processing temperature for graphitization can arouse nanoparticle ripening, agglomerations and compositional changes. Herein, we report a unique "popcorn-making-mimic" strategy for preparing a highly conjugated and uniformly compounded graphene@NiFe$_2$O$_4$ composite film through a laser-assisted instantaneous compounding method in ambient condition. It can successfully inhibit the unwanted structural disintegration and mass loss during the laser treatment by avoiding oxidation, bursting, and inhomogeneous heat accumulations, thus achieving a highly integrated composite structure with superior electrical conductivity and high saturated magnetization. Such a single-sided film exhibits an absolute shielding effectiveness of up to 20906 dB cm$^2$ g$^{-1}$ with 75% absorption rate, superior mechanical flexibility and excellent temperature/humidity aging reliability. These performance indexes signify a substantial advance in EMI absorption capability, fabrication universality, small form-factor and device reliability toward commercial applications. Our method provides a paradigm for fabricating sophisticated composite materials for versatile applications.

With the rapid development of Internet of Everything, smart communication terminals are more compact in their sizes, and thus crosstalk among the highly adjacent signal ports has become a critical challenge than ever[1,2]. Currently commercial metal-based electromagnetic interference (EMI) shielding films are adopted and patched around the transmission ports for casing electromagnetic waves by the reflection mechanism[3]. But as the circuits are being more intensively integrated, this mechanism becomes less effective. Such a situation would become

[1]Institute of Materials Research, Tsinghua Shenzhen International Graduate School, Tsinghua University, Shenzhen 518055, China. [2]Institute of Semiconductor Manufacturing Research, Shenzhen University, Shenzhen 518060, China. [3]Department of Materials Science and Engineering, University of Pennsylvania, Philadelphia, PA, USA. [4]Shenzhen Institute of Advanced Electronic Materials, Shenzhen Institute of Advanced Technology Chinese Academy of Sciences, Shenzhen 518055, China. [5]These authors contributed equally: Mingjie Liu, Zhiyuan Wang, Zhaoqiang Song. ✉e-mail: guozb@szu.edu.cn; yang.cheng@sz.tsinghua.edu.cn

even worse when the shielding films are burst or corroded after being repeatedly folded or aged in harsh conditions, where signal leaking takes place[4]. To overcome such a challenge, developing thin EMI shielding films with a strong electromagnetic wave absorption capability would be a more pronounced solution, which will substantially promote the development of the electronics and communications industry[5].

For this reason, graphene[6–8], carbon nanotubes[9,10], MXenes[11–14] and other low-dimensional, flexible, porous and structure-rich EMI shielding composite materials have received more attentions. These composites not only show excellent electrical conductivity for high EMI shielding ability, but can also provide a high absorption rate by virtue of their micro/nano- porous structures and rich composition[15]. Moreover, they have excellent overall properties such as thinness, high mechanical flexibility, corrosion resistance and ease of preparation[16,17]. Compounding magnetic nanometallic oxides with them can simultaneously elevate EMI shielding effectiveness and absorption rates, which is widely regarded as a promising strategy[18,19]. For example, Kumara et al. employed a strategy of nucleating and growing $NiFe_2O_4$ on the graphene oxide (GO) surface to fabricate $NiFe_2O_4$/rGO nanocomposite films by co-precipitation and hydrothermal methods, and obtained an EMI shielding level of 38.2 dB with 35% magnetic nanoparticle occupancy and the film thickness of 2 mm[20]. Huang et al. prepared $MXene/MWCNTs/SrFe_{12}O_{19}$ composite films (438 S cm$^{-1}$) by mixing the existing $SrFe_{12}O_{19}$ nano metal oxides with MXene and MWCNTs by vacuum filtration. Although the EMI shielding performance of 62.9 dB was obtained, the electric conductivity decreased by 80% compared with the original MXene (2210 S cm$^{-1}$)[21]. This can be ascribed to the lack of plethora porous structures and the limited chemical stability of MXene, which put on difficulties in compounding with magnetic particles when compared with carbon-based scaffolds. It can be seen that in-situ growth of nanoparticles on conductive scaffold surfaces and direct physical mixing are quite effective for preparing composite EMI shielding materials, but for further improvements in EMI shielding effectiveness and absorption rates, more universal and advanced methods are still lacking. From the perspective of fabrications, until now, the best way to enable good conductivity of monolithic structures based on graphene and graphene-like materials is either increasing the temperature or introducing a strong reducing agent to construct conjugated C = C bonds among the low-dimensional building blocks. But such conditions can easily cause agglomeration and denaturation of magnetic oxides by sintering or redox reactions[22,23]. The dilemma is, it is very difficult to form stable sp$^2$ hybridization between conducting building blocks for a highly conductive scaffold[24], in a compatible chemical condition where metal oxide nanomaterials can be formed and well-distributed in that scaffold without property degradation[25,26]. In addition, physically mixing the components is often invalid as the magnetic nanoparticles are very easily agglomerated. Therefore, new theoretical and methodological innovations are urgently required to achieve the multiple targets of high conductivity, low thickness, high EMI shielding effectiveness and absorption rate.

Laser processing methods are becoming a powerful toolbox for engineering various nanostructures in recent years, especially in precisely regulating materials shape, structure and composition[27,28]. Compared with the wet synthesis strategies such as hydrothermal and solvothermal methods, laser induced nanomaterials fabrication processes are simpler and greener. In addition, increasing the laser power and pulse oscillation rate can improve the fabrication throughput, thus making this method compatible to industrial manufacturing[29–32]. For example, Han et al. prepared the EMI shielding materials by embedding metal nanocrystals into the laser-induced graphene (LIG-NiFe) by laser scribing metal salt precursor solution-soaked cedar wood under argon atmosphere[33]. Regrettably, the blast effect of the laser pulse[34,35] and the dramatic volume expansion due to the pyrolysis of an organic

matter at instant high temperatures made it difficult to obtain an integrated film layer, thus limiting the scenarios for use. Yu et al. obtained a composite EMI shielding film of $Fe_3O_4$ nanoparticles-loaded LIG (LIG@$Fe_3O_4$) structure by laser processing a composite film of the mixture of metal salt precursor and polymer resin[36]. Even though the material showed a continuous film structure with a uniform loading of nanoparticles, since the LIG and $Fe_3O_4$ nanoparticles were generated simultaneously during the reaction of laser preparation, the $Fe_3O_4$ nanoparticles embedded in the LIG caused damage to the conductive network structure during the laser treatment as the square resistance of the composite film increased by ~100%[36]. The main reason for this is that the laser pulse has strong blast and thermal accumulation effects[29,32], which will inevitably bring about destructions to the conductive network structure. In particular, the simultaneous generation of nanoparticles will aggravate this damage[37,38]. Therefore, how to regulate the local laser energy distribution of heat in the material and to avoid the detrimental effects of bursting and heat accumulation is the key to solving the deterioration of the conductive network and the easy agglomeration of magnetic metal oxides in the materials compounding process. Unfortunately, to date, very few studies have been reported on regulating the laser heat distribution to obtain a more uniform size and loading of nanoparticles. Any of the technological breakthroughs will be extremely important to promote the development of both EMI shielding materials and other related areas.

Herein, we report a "popcorn-making" mimicking method to prepare the reduced graphene oxide (rGO) covered LIG composite film, which is homogeneously loaded with nickel ferrite nanoparticles (rGO/LIG@$NiFe_2O_4$), as a model for high quality electromagnetic wave absorption and shielding films. This method converts the local transient laser pulse energy into spatial uniformly-distributed heat by covering the LIG structure with a GO lid above. As a result, mass loss and structural damage caused by the local transient heat accumulation and high-temperature induced bursting are well-suppressed, and thus the damage to the conductive network of laser-induced graphene composite films is effectively avoided. This uniform thermal energy distribution enables a uniform distribution of magnetic nanometallic oxides with high crystallinity and ultrafine size (3.63 ± 0.98 nm). This composite film achieves a total EMI shielding effectiveness of 36 dB and 51 dB at 70 μm on one side and 166 μm on both sides, where the absorption ratios are enhanced to 75% and 73%, respectively. The significant improvements in EMI shielding effectiveness and absorption rate demonstrate that our "popcorn-making-mimic" method can provide a versatile and universal solution for developing advanced porous composite materials with superior performance characteristics in both materials distribution homogeneity and electrical conductivity.

## Results
### Preparations and morphological investigations of pristine LIG and rGO/LIG@$NiFe_2O_4$ composite films

As shown in Fig. 1A, B, here, GO is chosen as a protective lid not only because it is a low-cost and scalable material for preparing graphene but GO can be conveniently converted into conducting rGO[39]. What's more, prior researches have confirmed that GO is an effective flame retardant material, and thus the laser treatment process in ambient environment would hardly incur burning[40–42]. Specifically, during the reaction process of laser processing, the nearly insulating GO lid converts the local transient laser pulses energy into uniform thermal energy during the transition to rGO and confines it inside the LIG structure. This in turn allows the heat to uniformly diffuse inside LIG. Thus, our present method achieves effective utilization of the laser pulse energy by introducing a GO lid. This is analogous to the indispensable step of covering the pot when cooking the popcorn in kitchen, otherwise the corn flakes will spill out everywhere due to their drastic volume expansion. Correspondingly, it effectively prevents

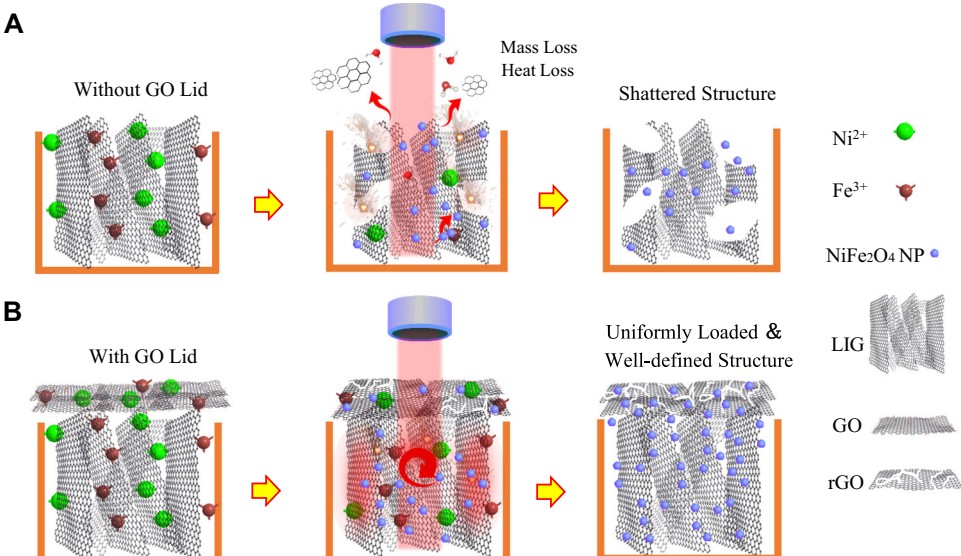

**Fig. 1 | Schematic illustration of different laser processing methods. A** A schematic showing the reaction process of Laser-induced graphene (LIG) in conventional laser-prepared composite films without Graphene oxide (GO) lid on the surface after introduction of metal salt precursors. **B** Reaction process for the laser preparation of composite films of LIG following the introduction of a metal salt precursor and a GO lid (The rGO/LIG@NiFe$_2$O$_4$ composite films prepared by the "popcorn-making-mimic" method).

violent combustion and uneven energy distribution, thus reducing the mass and energy loss during the drastic laser processing step.

As can be seen, the as-obtained LIG maintains a rich three-dimensional (3D) structure (Fig. 2A, B) with higher conductivity and crystallinity (Supplementary Fig. 1, 2). SEM images (Supplementary Fig. 3A) show that the thickness of LIG is about 71 μm, and the vertical 3D porous structure of the cross section is clearly visible. A comparison of Fig. 2C–F can indicate that the nanosheets of LIG were severely damaged without the GO lid. In addition, NiFe$_2$O$_4$ nanoparticles (Supplementary Fig. 2) were embedded in the LIG nanosheets (Fig. 2C, D). Figure 2E, F show that the structural integrity of LIG was well preserved by the GO lid. Figure 2G, H (also in Supplementary Fig. 17) and the corresponding elemental distribution show a homogeneous distribution of NiFe$_2$O$_4$ nanoparticles on LIG, resulting in the formation of rGO/LIG@NiFe$_2$O$_4$ composite films. The drastic morphological differences between rGO/LIG@NiFe$_2$O$_4$ and LIG@NiFe$_2$O$_4$ can be ascribed to the accumulation of heat and bursting effect caused by the laser pulse. As can be seen in Fig. 2I, the thickness of rGO/LIG@NiFe$_2$O$_4$ (F-N-0.7, "F" represents iron nitrate nonahydrate, "N" represents nickel chloride hexahydrate, and "X" represents the concentration of the composite films) is ~70 μm, which is almost the same as that of LIG (71 μm). Transmission electron microscopy images of rGO/LIG@NiFe$_2$O$_4$ (Fig. 2J–N) further show that the NiFe$_2$O$_4$ nanoparticles are uniformly distributed on LIG, with the average particle size of 3.63 ± 0.98 nm. Thereby, it is clear that we can effectively suppress the adverse effects caused by thermal accumulation and bursting in the conventional laser compounding process by involving the "popcorn-making-mimic" strategy.

## Structural analyses of LIG and rGO/LIG@NiFe$_2$O$_4$

The crystalline information of GO, LIG and different rGO/LIG@NiFe$_2$O$_4$ samples can be obtained through XRD analysis (Fig. 3A). From the XRD pattern of GO, the strong diffraction peak appearing at about 10.1° corresponds to the (001) crystal plane with the interlayer spacing of 0.9 nm[42]. LIG has two diffraction peaks at about 26.2° and 42.8°, corresponding to (002) and (100) crystal planes. Among them, the interlayer spacing of the (002) crystal plane is about 0.34 nm, which indicates the high degree of graphitization of LIG. The diffraction peak of LIG at about 42.8° is related to its rich 3D porous structure[27]. The interlayer spacing of GO is about three times of that of LIG, which is caused by the grafting of abundant functional groups such as epoxy groups, hydroxyl and carbonyl groups on the GO surface[43,44]. The diffraction peaks of different rGO/LIG@NiFe$_2$O$_4$ samples appear at about 18.4°, 30.3°, 35.7°, 37.3°, 43.4°, 53.9°, 57.4°, 63.0°, 71.5°, 74.6°, 75.6°, 26.2° and 42.8°. These diffraction peaks correspond to (111), (220), (311), (222), (400), (422), (511), (440), (620), (533), and (622) of NiFe$_2$O$_4$, and the (002) and (100) crystal planes of LIG and rGO respectively[45,46]. By comparing the XRD patterns of GO and different rGO/LIG@NiFe$_2$O$_4$ samples, it can be found that the diffraction peaks of GO have changed, which also indicates the conversion of GO to rGO. In addition, the sharp diffraction peaks in the XRD pattern of rGO/LIG@NiFe$_2$O$_4$ are completely consistent with the ICDD card no.01-089-4927 of NiFe$_2$O$_4$, which proves the high crystallinity and a pure phase of NiFe$_2$O$_4$[20]. The highly crystalline and uniformly distributed NiFe$_2$O$_4$ magnetic nanoparticles imply their superiority in electromagnetic wave absorption capability[47]. In addition, the effectiveness and significant advantages of this "popcorn-making-mimic" strategy for heat modulation are also well-confirmed.

The Raman spectra of the samples (Fig. 3B) can provide information about the structure and physical and chemical changes of the rGO/LIG@NiFe$_2$O$_4$ composite film. It can be seen from Fig. 3B that the typical D peak of LIG appears at 1350 cm$^{-1}$, which is attributed to the symmetrical stretching vibration of $sp^2$ carbon atoms, and is related to the defects and disorder of the material structure. In addition, the G peak (1580 cm$^{-1}$) and 2D peak (2700 cm$^{-1}$) are related to the in-plane stretching vibration of $sp^2$ carbon atoms and the stacking of graphene layers[48]. The D peak of GO is much sharper than that of other samples, which is attributed to the defects caused by the abundant functional groups such as epoxy groups, carbonyl and hydroxyl groups on its surface. In addition, from the Raman spectra of each rGO/LIG@NiFe$_2$O$_4$ composite film, it can be seen that the intensity of D band is decreased, indicating that with the transition from GO to rGO, the number of surface functional groups is decreased. Raman peaks (A$_{1g}$ + E$_g$ + 3T$_{2g}$) of NiFe$_2$O$_4$ (Supplementary Fig. 4) with inverse spinel structure are located at 688 cm$^{-1}$, 333 cm$^{-1}$, 202 cm$^{-1}$, 480 cm$^{-1}$ and 557 cm$^{-1}$, which are consistent with previous literature reports[20,49]. To note, the positions of the characteristic peaks of the Raman band of NiFe$_2$O$_4$ in different samples are almost the same, indicating uniform

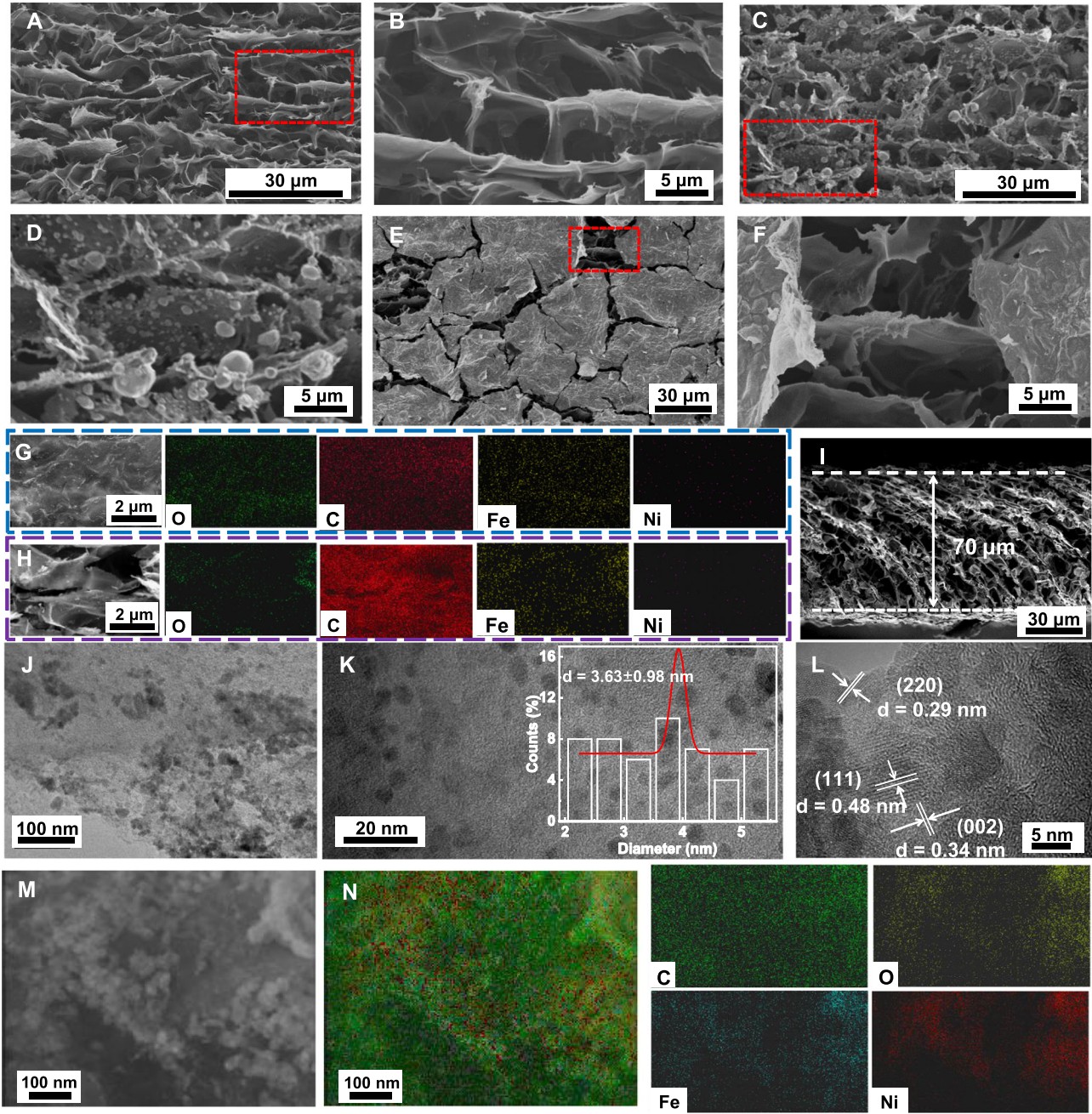

**Fig. 2 | Morphological characterizations of Laser-induced graphene (LIG), LIG@NiFe₂O₄ and rGO/LIG@NiFe₂O₄.** **A** SEM image of Laser-induced graphene (LIG) with low magnification (top-view). **B** High magnification SEM image of the red dashed line region in (**A**) (top-view). **C** SEM overview image of LIG@NiFe₂O₄ (top-view). **D** High magnification SEM image of the red dashed line region in (**C**) (top-view). **E** SEM overview image of rGO/LIG@NiFe₂O₄ (top-view). **F** High magnification SEM image of the red dashed line region in (**E**) (top-view). **G** SEM image of the upper surface of rGO. **H** SEM image of the internal LIG. EDS image of the upper surface and interior of rGO/LIG@NiFe₂O₄. **I** Cross-sectional SEM image of rGO/LIG@NiFe₂O₄. **J**–**L** TEM images of rGO/LIG@NiFe₂O₄ at different magnifications. The overlaid plot in K is a statistical result of the particle size of NiFe₂O₄ nanoparticles. **M** TEM image of rGO/LIG@NiFe₂O₄ composite film. **N** Images of overlapping and independent distributions of C, O, Fe and Ni in rGO/LIG@NiFe₂O₄ composite films.

structural characteristics. Figure 3C shows that the $I_D/I_G$ ratios of LIG, GO and different rGO/LIG@NiFe₂O₄ samples are about 0.60, 1.84, 0.84 (F-N-0.3), 1.02 (F-N-0.5), 1.08 (F-N-0.7), 1.19 (F-N-1), respectively. This also demonstrates the high degree of graphitization of LIG and the low degree of graphitization of GO[27,50]. The $I_D/I_G$ ratios of different rGO/LIG@NiFe₂O₄ samples are between LIG and GO, indicating that the loading of NiFe₂O₄ can influence the graphitization of LIG nanosheets[51]. Meanwhile, the graphitization degree of the composite film has increased. Figure 3D shows that the thickness of rGO/LIG@NiFe₂O₄ samples prepared under different metal salt precursor

concentrations are about 67 μm (F-N-0.3), 67 μm (F-N-0.5), 70 μm (F-N-0.7) and 73 μm (F-N-1). The gradual increase in the thickness of the rGO/LIG@NiFe₂O₄ composite films is mainly due to the reduction of GO to rGO and the loading of NiFe₂O₄ on rGO and LIG. In particular, as the concentration of the precursor salt increases, the loading of NiFe₂O₄ inside the composite film structure increases accordingly, leading to the corresponding increase in the thickness of the rGO/LIG@NiFe₂O₄ composite film (Supplementary Table 1).

In order to gain insights into the porous and structural information of LIG and different rGO/LIG@NiFe₂O₄ composite films, detailed

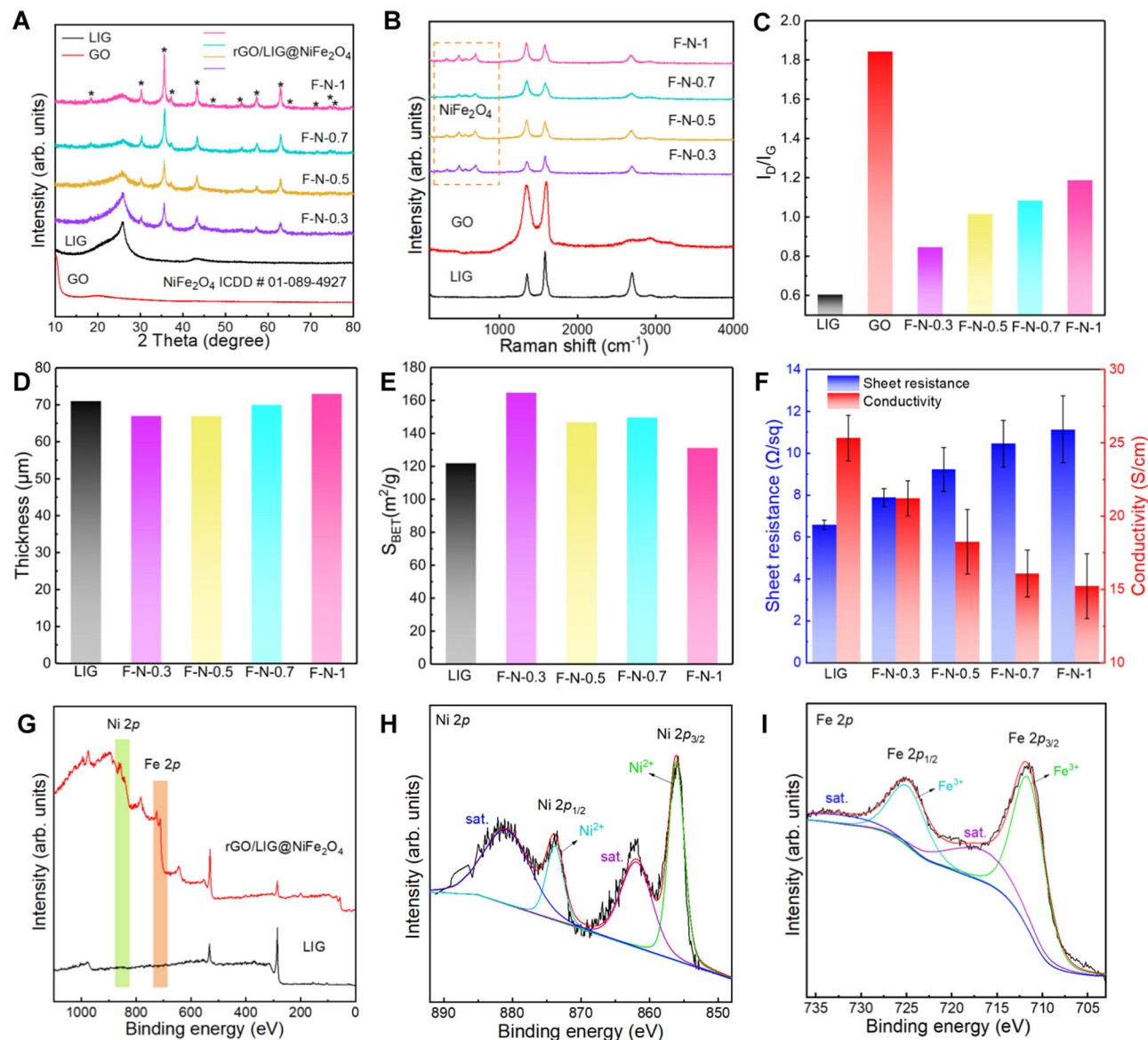

**Fig. 3 | Structural characterizations of different samples. A** XRD patterns of Graphene oxide (GO), Laser-induced graphene (LIG), and different rGO/LIG@NiFe$_2$O$_4$ samples. The peaks originating from NiFe$_2$O$_4$ are marked with *. **B** Raman spectra of the same samples. **C** I$_D$/I$_G$ ratios for GO, LIG and different rGO/LIG@NiFe$_2$O$_4$ samples. **D** Thickness of LIG and different rGO/LIG@NiFe$_2$O$_4$ samples. **E** Specific surface area of LIG and different rGO/LIG@NiFe$_2$O$_4$ samples. **F** Sheet resistance and conductivity of LIG and different rGO/LIG@NiFe$_2$O$_4$ samples. The error bars are derived from calculating the standard deviation of five samples. **G** XPS spectra of LIG and rGO/LIG@NiFe$_2$O$_4$. **H, I** High-resolution XPS spectra of (**H**) Ni(2$p$) and (**I**) Fe(2$p$) of rGO/LIG@NiFe$_2$O$_4$ composite films.

specific surface area information was obtained by the Brunauer Emmett Telle (BET) method, which is shown in Fig. 3E. The specific surface areas of LIG and rGO/LIG@NiFe$_2$O$_4$ composite films with different loadings of NiFe$_2$O$_4$ are 121.87 m$^2$ g$^{-1}$, 164.71 m$^2$ g$^{-1}$ (F-N-0.3), 146.69 m$^2$ g$^{-1}$ (F-N-0.5), 149.66 m$^2$ g$^{-1}$ (F-N-0.7) and 131.2 m$^2$ g$^{-1}$ (F-N-1) respectively. The large specific surface area of the LIG and rGO/LIG@NiFe$_2$O$_4$ composite films indicates their rich porous structure characteristics, which is important to the multiple reflection and absorption of electromagnetic waves inside the material. According to the specific surface area information, the rGO/LIG@NiFe$_2$O$_4$ composite films all exhibit ultra-high specific surface areas, which are higher than that of LIG. Given that the upper GO layer protects the highly crystalline LIG structure from damage by laser scribing, the specific surface area of LIG is unaffected. Furthermore, it is evident from the previous morphological and structural characterizations that NiFe$_2$O$_4$ is homogeneously deposited onto the surface of GO and inside the porous structure of LIG. In addition, the pore volume of rGO/

LIG@NiFe$_2$O$_4$ (Supplementary Fig. 5) is only slightly lower than that of LIG, further indicating the uniform loading of NiFe$_2$O$_4$ and the high integrity of the porous structure of LIG.

The electrically conductive property (Fig. 3F) is an important factor affecting the EMI shielding performance of the sample. The square resistance and electrical conductivity of LIG are 6.58 Ω/□ and 25.32 S cm$^{-1}$. The square resistances of different rGO/LIG@NiFe$_2$O$_4$ composite films are 7.88 Ω/□ (F-N-0.3), 9.23 Ω/□ (F-N-0.5), 10.47 Ω/□ (F-N-0.7), 11.13 Ω/□ (F-N-1), and their corresponding electrical conductivity are 21.20 S cm$^{-1}$, 18.25 S cm$^{-1}$, 16.08 S cm$^{-1}$, 15.39 S cm$^{-1}$. It can be seen that the electrical conductivity of the rGO/LIG@NiFe$_2$O$_4$ composite film is lower than LIG, which is mainly caused by the loading of NiFe$_2$O$_4$ on LIG. The decrease in conductivity of different rGO/LIG@NiFe$_2$O$_4$ composite films was 16% (F-N-0.3), 28% (F-N-0.5), 36.5% (F-N-0.7), and 39.2% (F-N-1), respectively, compared to that of LIG. The more NiFe$_2$O$_4$ loaded on the rGO/LIG@NiFe$_2$O$_4$ composite film, the more the electrical conductivity drops. In addition, the square

resistance and electrical conductivity of the LIG@NiFe$_2$O$_4$ composite films were 16.73 Ω/□ and 11.95 S cm$^{-1}$, respectively. Compared to LIG, its electrical conductivity decreased by 52.8%. This shows that although the electrical conductivity of the rGO/LIG@NiFe$_2$O$_4$ composite film is slightly lower than that of LIG, its electrical conductivity is significantly higher than LIG@NiFe$_2$O$_4$. This indicates that the presence of the rGO lid effectively protects the conductive network of the substrate compared with the conventional direct laser processing method.

The LIG and rGO/LIG@NiFe$_2$O$_4$ composite films were characterized and analyzed by X-ray photoelectron spectroscopy (XPS) to obtain a comprehensive understanding of structure and chemical composition information. The C 1 s and O 1 s characteristic peaks of LIG and their corresponding binding energies, as well as the C 1 s, O 1 s, Fe 2$p$ and Ni 2$p$ peaks of the rGO/LIG@NiFe$_2$O$_4$ composite films and their corresponding binding energies are provided (Fig. 3G). By comparing the XPS spectra of LIG and rGO/LIG@NiFe$_2$O$_4$, it can be concluded that NiFe$_2$O$_4$ was successfully loaded onto rGO and LIG. The two characteristic peaks in the high-resolution XPS spectrum of Ni 2$p$ at 856.1 eV and 873.8 eV correspond to Ni 2$p_{3/2}$ and Ni 2$p_{1/2}$, respectively (Fig. 3H). The satellite peaks of Ni 2$p_{3/2}$ and Ni 2$p_{1/2}$ of Ni$^{2+}$ appear at 861.9 eV and 880.8 eV, indicating that the valence state of Ni in rGO/LIG@NiFe$_2$O$_4$ composite film is Ni$^{2+}$, and no other valence state exists. The high-resolution XPS spectrum of Fe 2p shows two characteristic peaks corresponding to Fe 2$p_{3/2}$ and Fe 2$p_{1/2}$ at 711.5 eV and 724.9 eV, respectively (Fig. 3I). At the same time, the two satellite peaks of Fe 2$p_{3/2}$ and Fe 2$p_{1/2}$ appear at 716.6 eV and 732.1 eV, indicating that the valence state of Fe in the rGO/LIG@NiFe$_2$O$_4$ composite film is Fe$^{3+}$ [20]. High-resolution XPS spectra of C 1 s and O 1 s of LIG and rGO/LIG@NiFe$_2$O$_4$ are also given (Supplementary Fig. 6). The three peaks appearing at 284.8 eV, 285.1 eV and 288.5 eV in the high-resolution XPS spectrum of LIG C 1 s (Supplementary Fig. 6A) correspond to C-C, C = C and O-C = O, of which C-C is dominant. Compared with the high-resolution XPS spectrum of the C 1 s of LIG, the C-O appears at 286.1 eV, which is due to the formation of NiFe$_2$O$_4$ (Supplementary Fig. 6C). The two peaks in the high-resolution XPS spectrum of LIG O 1 s (Supplementary Fig. 6B) appearing at 532.4 eV and 533.6 eV correspond to the C-O-C and O = C-O bonds. Compared with the high-resolution XPS spectra of LIG O 1 s, rGO/LIG@NiFe$_2$O$_4$ (Supplementary Fig. 6D) has an extra peak at 530.3 eV, corresponding to that of NiFe$_2$O$_4$ [46,52].

LIG has excellent hydrophilicity, and its water contact angle is only 17° (in air atmosphere), which guarantees the sufficient infiltration of the precursor salt solution. After loading NiFe$_2$O$_4$ nanoparticles, the water contact angles of LIG@NiFe$_2$O$_4$ and rGO/LIG@NiFe$_2$O$_4$ composite films change to 20.7° and 53.5°, respectively (Supplementary Fig. 7). This shows that the loading of NiFe$_2$O$_4$ nanoparticles improves the hydrophobicity. In addition, due to the thermal reduction of oxygen-containing functional groups after laser treatment, the surface of rGO/LIG@NiFe$_2$O$_4$ composite films are more hydrophobic compared to LIG@NiFe$_2$O$_4$. For the electromagnetic shielding materials to be applied to devices and electronic devices in the future, the better hydrophobicity is favorable for the protection of the devices.

In order to explore the thermal stability of LIG and LIG loaded with magnetic metal oxide nanoparticles, as well as the influence of different loading levels of nanoparticles on the thermal decomposition process of LIG in the air, we performed TGA-DSC and the curves of the samples were obtained (see Supplementary Fig. 8). It can be seen that the introduction of NiFe$_2$O$_4$ nanoparticles can promote the combustion process of LIG where the onset temperature of LIG pyrolysis is down-shifted. In comparison, with the introduction of the fire-retardant rGO lid (Supplementary Fig. 9), the combustion process is retarded and the TGA peak corresponding to the LIG pyrolysis gets smoother, where the exothermic peak splits into two, which means that the more uniform and smaller NiFe$_2$O$_4$ nanoparticles accelerate the pyrolytic process of the LIG fragment which they attach to [53,54], and

the pyrolysis process becomes more thermal-stable with the presence of the rGO lid.

## Magnetic and EMI shielding properties

In order to investigate the magnetic loss mechanism of rGO/LIG@NiFe$_2$O$_4$ composite films, we tested the hysteresis loops at room temperature of the rGO/LIG@NiFe$_2$O$_4$ composite films prepared with different precursor salt concentrations (Fig. 4A). The saturation magnetization intensities of different rGO/LIG@NiFe$_2$O$_4$ composite films are 1.88 (F-N-0.3), 2.84 (F-N-0.5), 3.32 (F-N-0.7) and 4.60 (F-N-1) emu g$^{-1}$, respectively. The increase of saturation magnetization with the increase of the precursor salt concentration involved in the reaction is related to the gradual increase of the relative content of NiFe$_2$O$_4$ in the rGO/LIG@NiFe$_2$O$_4$ composite film. It can be seen more clearly through the partially enlarged view of the room temperature hysteresis loop (Fig. 4B) that the coercivities of different rGO/LIG@NiFe$_2$O$_4$ composite films were 117 Oe (F-N-0.3), 119 Oe (F-N-0.5), 126 Oe (F-N-0.7) and 129 Oe (F-N-1). From the low coercivity, NiFe$_2$O$_4$ exhibits the typical soft magnetic characteristic of easy magnetization. Such a characteristic means that a larger magnetization intensity can be achieved with a smaller external magnetic field. In addition, this feature can effectively improve impedance matching and attenuate electromagnetic waves through magnetic losses [55,56].

The EMI shielding effectiveness of LIG and rGO/LIG@NiFe$_2$O$_4$ composite films were evaluated in detail. Figure 4C shows the total EMI shielding effectiveness (SE$_T$) of LIG and different rGO/LIG@NiFe$_2$O$_4$ composite films at 8.2–12.4 GHz (X band), where the highest SE$_T$ of LIG is 25.2 dB. When the metal salt concentration of the precursor was 0.3, 0.5, 0.7 and 1.0 M, the highest SE$_T$ of the rGO/LIG@NiFe$_2$O$_4$ composite film was 29.5 dB (F-N-0.3), 34.3 dB (F-N-0.5), 36.0 dB (F-N-0.7) and 33.3 dB (F-N-1), respectively. It can be seen that, compared with LIG, the introduction of NiFe$_2$O$_4$ significantly improves the overall EMI shielding effectiveness of the film. Notably, when the metal salt concentration of the precursor increases from 0.7 M to 1 M, the total EMI shielding effectiveness begins to decrease. A plausible explanation is that the excessive loading of NiFe$_2$O$_4$ adversely affects the conductive network of the composite film [36], thus weakening the contribution of the ohmic loss from the conductive network to the EMI shielding performance. In addition, in order to more objectively evaluate the EMI shielding effect of the sample films in the entire X-band, the average EMI shielding performance was calculated (Fig. 4G). The average SE$_T$ of LIG in the X-band is 23.8 dB. The average SE$_T$ values of the composite films are 27.8 (F-N-0.3), 32.6 (F-N-0.5), 34.4 (F-N-0.7) and 31.8 (F-N-1) dB. Compared to LIG, the composite films show better EMI shielding effectiveness throughout the X-band. The total EMI shielding effectiveness (SE$_T$) is composed of absorption (SE$_A$), reflection (SE$_R$) and internal multiple reflection (SE$_{MR}$), and when the absorption (SE$_A$) exceeds 10 dB, the internal multiple reflection (SE$_{MR}$) will be counted as absorption (SE$_A$) [57]. In order to further understand the contribution of each part to the total EMI shielding effectiveness, SE$_R$ and SE$_A$ are given here (Fig. 4D, E). It can be seen from Fig. 4D, E that the increase in the SE$_T$ of the rGO/LIG@NiFe$_2$O$_4$ composite film is mainly ascribed to the increase in the absorption (SE$_A$). As can be seen from Fig. 4E, the maximum SE$_A$ of LIG is only 15.5 dB, while the maximum SE$_A$ of rGO/LIG@NiFe$_2$O$_4$ composite film reaches 25.6 dB when the concentration of the precursor metal salt is 0.7 M. This is attributed to the homogeneous loading of NiFe$_2$O$_4$ inside LIG and the integrity of the 3D conductive network structure of LIG. In addition, the average SE$_R$ of the rGO/LIG@NiFe$_2$O$_4$ composite film is only slightly increased (<10.5 dB) compared to the average SE$_R$ of LIG (8.6 dB). The main reason is that loading NiFe$_2$O$_4$ improves the impedance matching performance of the rGO/LIG@NiFe$_2$O$_4$ composite film, which significantly facilitates the electromagnetic waves entering the shielding material and also triggers a slight increase in SE$_R$ [58]. In addition, Fig. 4F reveals the change in the absorption rate of LIG and rGO/LIG@NiFe$_2$O$_4$

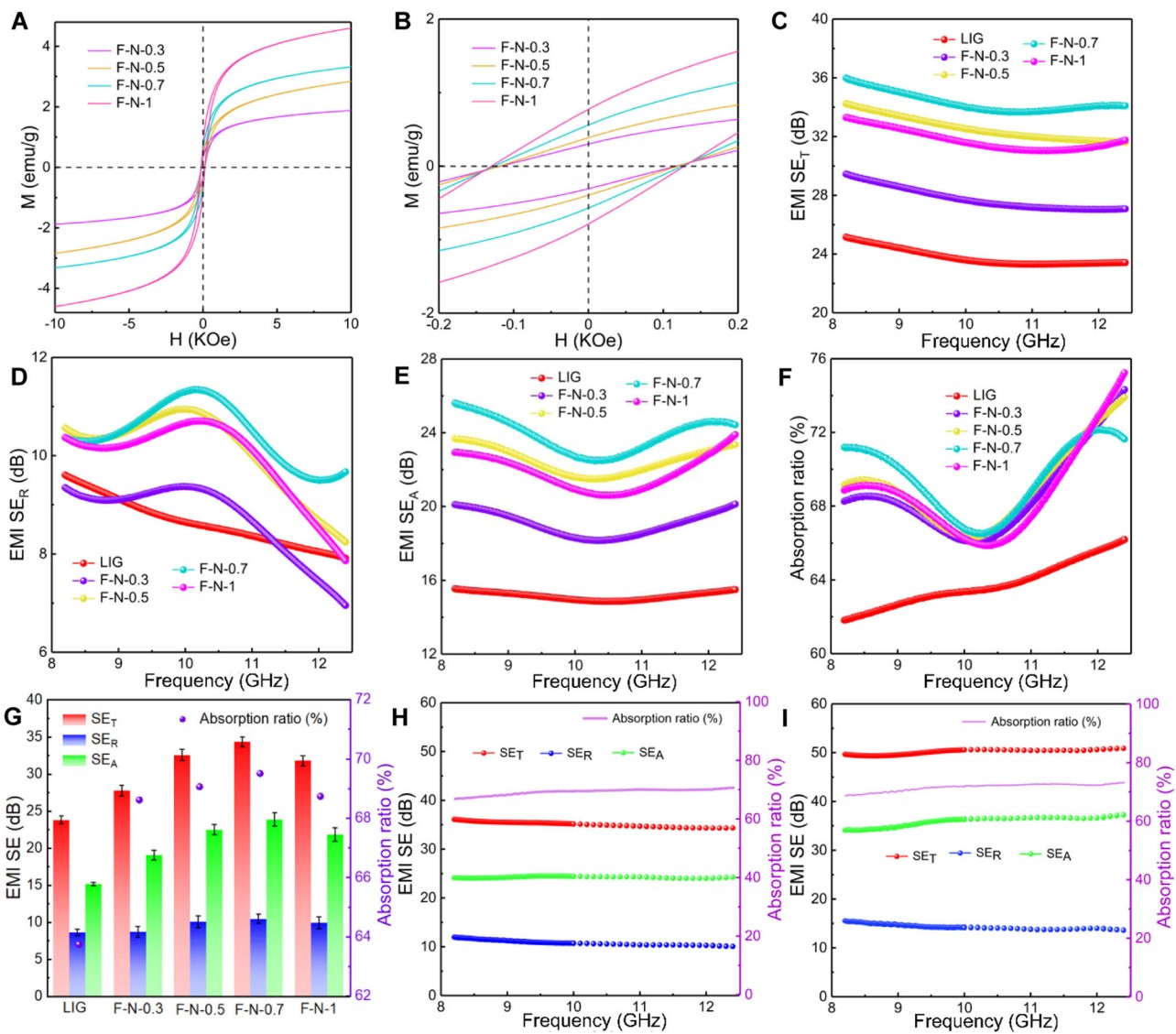

**Fig. 4 | Magnetic and EMI shielding effectiveness characterization.**
**A**, **B** Hysteresis loop and partially enlarged view of the rGO/LIG@NiFe$_2$O$_4$ prepared under different precursor salt concentrations. **C**–**E** EMI SE$_T$, SE$_R$ and SE$_A$ of Laser-induced graphene (LIG) and different rGO/LIG@NiFe$_2$O$_4$ samples in X band. **F** Absorption ratios of LIG and different rGO/LIG@NiFe$_2$O$_4$ samples at X band.

**G** The average EMI SE$_R$, SE$_A$, SE$_T$ and absorption ratios of LIG and different rGO/LIG@NiFe$_2$O$_4$ samples in X band. The error bars are derived from calculating the standard deviation of 201 points in the X-band. **H**, **I** EMI SE$_T$, SE$_R$, SE$_A$ and absorption ratios of double-sided LIG and double-sided rGO/LIG@NiFe$_2$O$_4$ (F-N-0.7) samples in X band.

composite films in the entire X band. It can be seen that the maximum absorption rate of LIG is 66.2%, and the maximum absorption rate of rGO/LIG@NiFe$_2$O$_4$ composite films is 75.2%. Figure 4G reveals that the average absorption efficiency of LIG and rGO/LIG@NiFe$_2$O$_4$ composite films in the entire X-band is 63.7% and 69.5%. Compared to LIG, the rGO/LIG@NiFe$_2$O$_4$ composite film has an improved absorption ratio throughout the X-band, rather than being limited to a specific frequency. Figure 4H, I further show that the optimal EMI SE$_T$ for the double-sided LIG and double-sided rGO/LIG@NiFe$_2$O$_4$ (F-N-0.7) in X-band are 36 dB and 51 dB, and the highest absorption rates are 70.7% and 73.3%. Compared to the single-sided films, double-sided films exhibit significantly higher and more stable EMI shielding effectiveness and absorption rates throughout the X-band, due to the higher film thickness and more loaded magnetic metal oxide. The EMI SE$_T$ and SE$_A$ of the double-sided rGO/LIG@NiFe$_2$O$_4$ (F-N-0.7) are 15 and 11 dB higher than the double-sided LIG, respectively. Finally, the SE$_T$, SE$_A$, SE$_R$ and absorption rates of the LIG@NiFe$_2$O$_4$ composite films produced by the conventional direct laser induction were 27.6 dB, 17.4 dB, 10.2 dB and 63% (Supplementary Fig. 10). It can be seen that the EMI shielding

effectiveness and absorption are only merely improved, because without the GO lid protection, the conductive network suffers severe damage under the intense laser energy onset and the metal oxide suffers a mass loss due to violent bursting and gas expansion. This demonstrates the significant advantage of our "popcorn-making-mimic" strategy in obtaining composite films with both high EMI shielding effect and high absorption rate.

## Mechanical properties and reliability of LIG and rGO/LIG@NiFe$_2$O$_4$ based EMI shielding films

Many of the materials that have been reported to have good EMI shielding properties in the past have had their mechanical properties and stability which are not clearly disclosed. As the complexity of future EMI shielding scenarios steadily increases, the mechanical properties and stability of the material become particularly important. The rGO/LIG@NiFe$_2$O$_4$ composite film is slim and flexible, allowing for reshaping or bending into desirable structures (Supplementary Fig. 11). The stress-strain curves of LIG and rGO/LIG@NiFe$_2$O$_4$ (F-N-0.7) showed that the tensile strength of the composite films was increased

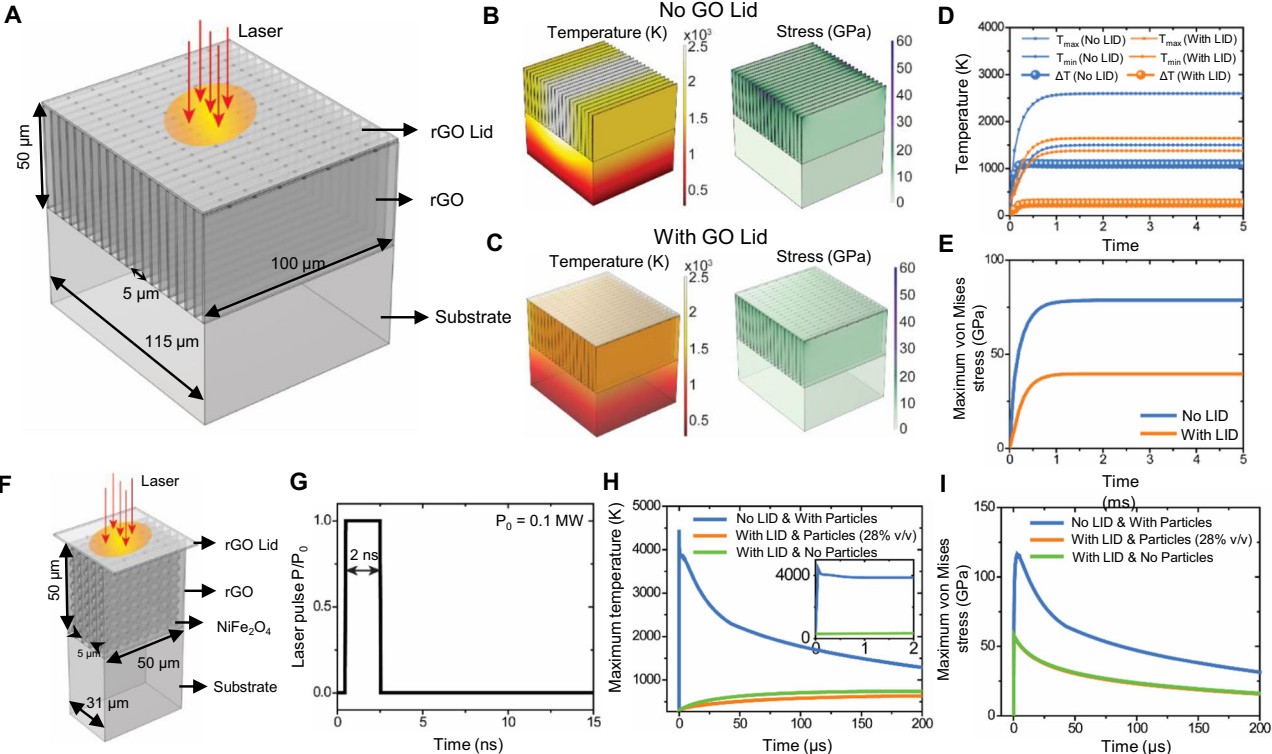

**Fig. 5 | Finite element analysis of the rGO/LIG@NiFe₂O₄ and LIG@NiFe₂O₄ composites during laser processing. A** The rGO/ LIG@NiFe₂O₄ composite being subjected to 1 s of heating from a 2.4 W laser. The yellowish shadow illustrates the laser spot, which directly heats the rGO lid (depicted in semitransparent light gray). The red arrow indicates the direction of the laser light. **B**, **C** The temperature and stress distribution in LIG@NiFe₂O₄ composite and rGO/LIG@NiFe₂O₄ composite.

**D**, **E** The maximum temperature and maximum von Mises stress in rGO layers (excluded the lid) in LIG@NiFe₂O₄ composite and rGO/LIG@NiFe₂O₄ composite. **F**, **G** The effect of pulsed power with power duration 2 ns on the failure of rGO. **H**, **I** The maximum temperature and maximum von Mises stress in rGO layers (excluded the lid) in LIG@NiFe₂O₄ composite, rGO/LIG@NiFe₂O₄ composite, and rGO/LIG. In (**I**), the orange and green lines are nearly overlapped.

by 8.4 MPa after loading the magnetic nanoparticles, while the elongation at break of the composite films and LIG were quite alike, both being > 20% (Supplementary Fig. 12A, B). This is because with the GO lid protection, the mass retention of the carbon-containing species is increased (with the GO protection the mass retention is improved over 10% of the original weight after the laser treatment, see Supplementary Table 2), i.e., the residual carbon rate is increased, and therefore the integrity of the conductive network and the mechanical structure are enhanced[59], while we only introduced a thin layer of GO lid (thickness ~ 2 μm, as can be seen in Supplementary Table 2 and its corresponding analysis). In addition, after scrapping the LIG off from the PI surface, the stress-strain curve of the PI shows that it has a tensile strength of up to 90 MPa, which provides the basis for the high mechanical strength of the composite film. However, its maximum elongation at break is only 13% (Supplementary Fig. 13), this indicates that its structural integrity is damaged under the laser treatment. After 10,000 bending cycles with a bending diameter of ~2 cm (Supplementary Fig. 14), the changes in square resistance for LIG and rGO/LIG@NiFe₂O₄ (F-N-0.7) samples were -16.4% and 2.6% respectively. This is attributed to the more integral structure of the LIG in the presence of the GO lid and the finer and more uniformly distributed size of the NiFe₂O₄, which facilitates to spread out the stress in the local areas (Supplementary Fig. 12). In addition, after 500 h of aging at 85 °C and 85% relative humidity, the square resistance change rates of LIG and rGO/LIG@NiFe₂O₄ were 8.0% and 2.3% (Supplementary Fig. 12C). Both LIG and rGO/LIG@NiFe₂O₄ have excellent reliability, but the stability of the composite film is more excellent. This is because that the loading of NiFe₂O₄ nanoparticles and the surface layer of rGO provide a good protection to the composite film. The EMI shielding performance of the LIG was greatly improved after bending and aging, reaching to

about 32.5 dB (Supplementary Fig. 12E, F). This anomaly is similar to a previous report, and a reasonable explanation is that the bending and aging treatments are disadvantageous to maintaining the regular honeycomb structure of the LIG, thus altering the electric conductivity and EMI shielding effectiveness of the material[12]. However, the EMI shielding performance and absorption rate of the rGO/LIG@NiFe₂O₄ composite film remained basically unchanged before and after bending, which is consistent with the change of electrical conductivity, further indicating the excellent stability of the composite film (Supplementary Fig. 12F).

## GO lid mechanism analysis
To further elaborate on the role of the GO lid in regulating the heat distribution during laser processing, we developed a container model as illustrated in the schematic diagram (Fig. 5A) using finite element modeling. Specifically, the rGO/LIG@NiFe₂O₄ and LIG@NiFe₂O₄ composites undergo 1 s of heating from a 2.4 W laser. As shown in Fig. 5B, C, the rGO/LIG@NiFe₂O₄ composites exhibit considerably lower temperatures and stress levels in the rGO layers (excluding the lid) compared to the LIG@NiFe₂O₄ composite. This discrepancy can be attributed to the role of the rGO lid, which functions as a thermal insulator, effectively shielding and protecting the underlying rGO layers. The protective function of the rGO lid is further emphasized as it mitigates temperature and stress in the underneath rGO layers. To illustrate the temporal evolution of these parameters, Fig. 5D, E demonstrate the kinetics of the temperatures and maximum vons Mises stress within the rGO layers. Significantly, both temperature and stress reach a steady state within a remarkably short duration of 1 millisecond. The thermal stress in the rGO layers, without the GO lid protection, approaches the theoretical strength limit (130 GPa,

Supplementary Table 3). This condition promotes the development of defects, including flaws or microcracks. These defects can then coalesce to form continuous cracks that fragment the rGO layers, resulting in mass loss. Meanwhile, Fig. 5C also illustrates that the rGO lid homogenizes the temperature of the large-area underlying rGO layers.

In practical scenarios, the laser operates in a pulsed mode, with a power duration <1 millisecond. We performed simulations to analyze heat transfer and stress variation, taking into account the pulsed laser power with a duration of 2 nanoseconds. In the case of pulsed laser (Fig. 5F, G), the intermittent power of the laser triggers elevated temperature and stress levels during the heating process in the LIG@NiFe$_2$O$_4$ composite. In comparison, the temperature within the substrate rGO layer of the rGO/LIG@NiFe$_2$O$_4$ composite is notably lower (Fig. 5H). This discrepancy arises from the distinct thermodynamics in the two composites. In the rGO/LIG@NiFe$_2$O$_4$ composite, energy is stored in the rGO lid, which subsequently functions as a continuous heat source for the substrate rGO layer. Consequently, the rGO layer experiences a more controlled and moderated temperature increase in contrast to the LIG@NiFe$_2$O$_4$ composite. This observation underscores the significant role of the rGO lid in regulating and distributing thermal energy, contributing to a more controlled thermal response in the composite. Notably, the 28% v/v is estimated from our experimental data, and as a matter of fact, the NiFe2O4 particles have a very slight contribution of this thermodynamic process (Fig. 5I).

In addition, the difference in thermal distribution between the two systems (the rGO/LIG@NiFe$_2$O$_4$ composite and the LIG@NiFe$_2$O$_4$ composite) directly affects the nucleation and growth process of nanoparticles. Generally, an increase in sintering temperature elevates the energy available to the particles, thereby enhancing the mobility of atoms or ions on their surfaces and leading to a higher diffusion rate. Consequently, higher temperatures accelerate the rate at which particles coalesce and grow, resulting in larger particle sizes within the composite. Regarding laser heating, the effect of laser power on particle size can be referred to the empirical formula: $D = K(P/v)^a$ ($D$ is the crystallite dimension, K and a are constants related to the precursor salts, $P$ is the laser power, and $v$ is the laser scanning speed)[60,61]. Without introducing the GO lid, the energy of the transient high-energy laser is injected into the system directly, resulting in an uneven heat distribution in the LIG. Consequently, the substantial temperature gradient causes pronounced disparities in particle size. Therefore, achieving a uniform formation of sub-10 nm nanoparticles within a 3D porous structure using transient lasers has been a formidable task, as it requires a stable and sufficiently low input power ($P$) coupled with an ultra-high laser scanning speed ($v$). In this study, the introduction of a GO lid addresses this issue by transforming the energy input from a localized, intermittent process to a sustained, widespread one, akin to achieving a high laser scanning speed. This approach not only enhances the retention rate of the species by converting concentrated laser power into a more diffuse, stable input over a larger area but also results in the production of nanoparticles with a more uniform sub-10 nm size distribution. Therefore, our method provides a paradigm in the regulation of the microsystem environment of the reaction process and the preparation of nanomaterials, setting a new standard in the field.

### Evaluation of the EMI shielding performance
In order to evaluate the EMI shielding performance of the composite films more objectively and comprehensively, we compared the EMI shielding performance and absorption rate of some carbon-based and MXenes-based composite EMI shielding materials with different thicknesses reported in recent years. Although there are different testing conditions and comparison dimensions for the comparison of EMI shielding performance, the comparison of the two core dimensions of thickness and EMI shielding effectiveness can reflect the differences between individual materials. As shown in Fig. 6A (also in

Supplementary Table 5), rGO/LIG@NiFe$_2$O$_4$ achieves an ultra-high EMI shielding effectiveness of 51 dB and 73% absorption at a thickness of 166 μm (Supplementary Fig. 15). This composite film not only has significantly better EMI shielding performance and absorption rate than other carbon-based EMI shielding composite materials, but also has an excellent overall performance such as high processing efficiency, low mass density, high mechanical flexibility and temperature-humidity stability. In order to further demonstrate its advantages regarding the future critical performance indexes for EMI shielding uses, a six-dimensional radar plot is presented here (Fig. 6B, detailed data in Supplementary Table 4). It can be seen that rGO/LIG@NiFe$_2$O$_4$ comprehensively performs well in these dimensions compared with most reported EMI shielding materials such as carbon-based, metal-based, and MXenes-based ones. The excellent EMI shielding performance and absorption rates originate primarily from the integrity of the 3D porous conductive network of LIG and the uniform loading of NiFe$_2$O$_4$. The rGO@NiFe$_2$O$_4$ layer will improve the impedance matching performance and cause partial electromagnetic wave signal losses. The electromagnetic waves entering the rGO@LIG@NiFe$_2$O$_4$ structure will be attenuated by the ohmic loss caused by the highly conductive LIG, the multiple reflections and absorption in the rich porous structure and defects inside the rGO/LIG@NiFe$_2$O$_4$, and the magnetic loss caused by the NiFe$_2$O$_4$ nanoparticles (Fig. 6C)[47,62]. By simply engineering the rGO/LIG@NiFe$_2$O$_4$ composite on both sides of a film, the EMI shielding effectiveness can be further enhanced. This evidence collectively shows that the integrity of the conductive network of the conductive substrate and the uniform loading of nanoparticles are essential to enhance EMI shielding performance and absorption ratio.

## Discussion
In summary, we demonstrated a "popcorn-making-mimic" strategy for preparing a highly conductive and porous composite film, which maximizes the magnetoelectric synergistic effect by uniformly loading stable magnetic nanoparticles inside the conductive and porous nanocarbon host. This method drastically improves the laser pulse energy deliver efficiency, overcoming the long-existing challenge of inhomogeneous heat distribution and avoiding the blast and combustion effects. It achieves a total EMI shielding effectiveness of 36 dB at 70 μm (on one side) and 51 dB at 166 μm (on double sides) on a PI substrate, with the corresponding absorption ratios enhanced to 75% and 73%, respectively. In addition, the rGO/LIG@NiFe$_2$O$_4$ composite film achieves an ultra-high absolute shielding effectiveness (SSE/t) of 20906 dB cm$^2$ g$^{-1}$ and a tensile strength up to 57.5 MPa. It can withstand even 10000 times of bending and 500 h of harsh aging without notable property degradation.

Our strategy here not only provides insights into the development of EMI shielding mechanisms and materials, it also shows excellent fabrication universality, small form-factor and device reliability. Such composite structures can not only be prepared on a large scale by increasing the laser power output and beam size, but their properties can be modulated by adjusting the precursors and processing parameters conveniently. It opens a new paradigm for compounding sophisticated nanostructures for versatile advanced applications.

## Methods
### Materials
Commercial polyimide (PI) films (Thickness:125 μm) were obtained from Shenzhen Changdasheng Electronics Co., Ltd (China). Nickel chloride Hexahydrate (NiCl$_2$·6H$_2$O) and ferric nitrate nonahydrate (Fe(NO$_3$)$_3$·9H$_2$O) were purchased from Shanghai Maclean Biochemical Technology Co., Ltd. Graphene oxide (percentage of single layer ≥ 99.5%, the average diameter range was 5–8 μm, and the concentration was 10 mg g$^{-1}$) was obtained from Gaoxi Technology Co., Ltd. (Hangzhou, China). All other chemicals were of analytical grade and used without further purification.

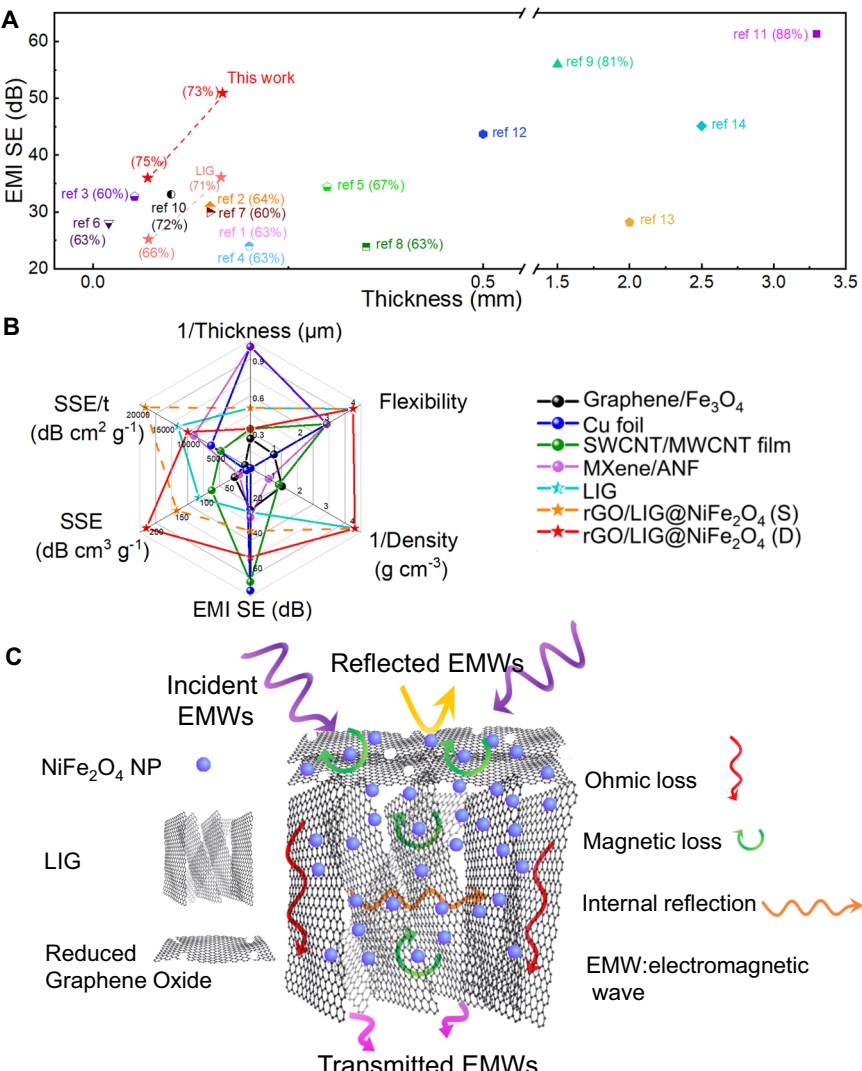

**Fig. 6 | Evaluation of the EMI shielding performance and a schematic of plausible mechanism. A** Comparison of EMI shielding effectiveness and absorption rates of Laser-induced graphene (LIG) and rGO/LIG@NiFe$_2$O$_4$ with other reported EMI shielding materials. The materials in the top left corner imply higher EMI shielding performance and thinner thicknesses. **B** A radar plot showing the comparison of the comprehensive performance of LIG and rGO/LIG@NiFe$_2$O$_4$ with other recently-reported representative EMI shielding materials. **C** A schematic image comprehensively showing the contribution of each component in EMI shielding mechanism of the rGO/LIG@NiFe$_2$O$_4$ composite film.

## Fabrication of LIG films

LIG was produced by one-step direct laser writing of polyimide films. Specifically, the polyimide film with the size of 10 × 15 cm was cleaned by ethanol and deionized water in turn, and then it was flatly bonded to a clean glass substrate with copper tape for laser scribing. A fiber laser with the wavelength of 1064 nm was used to prepare a highly conductive graphene film with the parameters of a defocus beam size of 10 mm, a scanning speed of 50 mm s$^{-1}$, a scanning pitch of 0.01 mm, and a power percentage of 15% (The power was 4.22 W, which was measured with a power meter).

## Fabrication of rGO/LIG@NiFe$_2$O$_4$ and LIG@NiFe$_2$O$_4$ composite films

The laser-induced graphene film prepared in the previous step was used as a substrate, and then 150 μL of nickel chloride hexahydrate aqueous solution (The specific concentration was 0.3 M, 0.5 M, 0.7 M and 1.0 M) was uniformly dropped on the graphene plane with a size of 4 × 4 cm and dried in an oven at 60 °C for 1 h. Next, 2.0 g of GO aqueous dispersion (with a concentration of 10 mg g$^{-1}$) and 10 mL ferric nitrate nonahydrate solution with the same concentration as nickel chloride

hexahydrate were mixed and stirred evenly. Then, A mixture of 300 μL GO and ferric nitrate nonahydrate were drop-casted on the dried graphene film containing nickel chloride hexahydrate, and then placed in an oven at 60 °C for 1 h. Notably, a simple step of drop-coating all three components can also be employed which can lead to the same result; yet splitting into two steps can facilitate optical microscopic observations to eliminate hollowing in the LIG area after the first step. The above-mentioned dried graphene-loaded iron salt, nickel salt and GO composite film was placed in a fiber laser processing platform and processed in ambient. The composite film was processed under the laser parameters with a defocus of 10 mm, a scanning speed of 250 mm s$^{-1}$, a scanning pitch of 0.005 mm, and a power percentage of 6% (The power was 2.43 W, Supplementary Fig. 16 shows the exploration of suitable processing parameters). In addition, the samples prepared with different concentrations of ferric nitrate nonahydrate and nickel chloride hexahydrate were labeled "F-N-X", where "F" represents iron nitrate nonahydrate, "N" represents nickel chloride hexahydrate, and "X" represents the concentration of the composite films, all concentrations were in the unit of M. The LIG@NiFe$_2$O$_4$ composite films were processed with the same parameters as rGO/LIG@NiFe$_2$O$_4$

(F-N-0.7), the only difference being that no GO dispersion was added to the precursor salt solution.

## Characterization

The morphological information of the samples was analyzed by field emission scanning electron microscope (FE-SEM, SAPPHIRE SUPRA 55). The crystallographic information of the sample was characterized by X-ray diffraction (XRD, Bruker DSRINT2000/PC, Germany) (Scanning rate was $5°\,min^{-1}$, scanning angle ranged from 10° to 80°). The square resistance and electrical conductivity of the thin film samples were measured by a four-probe resistance analyzer (MCP-T610, LARESTA-GP, Japan). The Raman spectrum of the sample was measured by a micro laser confocal Raman spectrometer (Horiba LabRAM HR800). X-ray photoelectron spectroscopy (XPS) was performed on the sample using Al-Ka radiation (50 W, 15 kV) (ESCAABSB 250Xi). FEI Tecnai G2 F30 was used to obtain high-resolution transmission electron microscopy (HRTEM) images of the sample. The magnetic information of the samples was measured by a vibrating sample magnetometer (VSM, bkt-4500z). The EMI shielding effectiveness information of the samples in the X-band (8.2-12.4 GHz) was obtained by first measuring the scattering parameters ($S_{11}$, $S_{21}$, $S_{22}$ and $S_{12}$) with a vector network analyzer (VNA, PNA-N5244A, Agilent), and then EMI SE was calculated based on the scattering parameters. The mass density of LIG sample was obtained by dividing the mass by the volume. A rectangular graphene film with a fixed area (S) was cut by laser, and then the volume ($V = S·H$) was calculated based on the thickness information (H) shown by the graphene cross-section image. The mass of the rectangular graphene film sheet (M1) was weighed on a balance, then the graphene on the surface was peeled off and weighed again to obtain the mass M2, and thus giving the mass of graphene as M1−M2. Finally the density could be obtained as $\rho = \frac{M1-M2}{S\cdot H}$. The calculation for the composite film was the same as above. The average size of the nanoparticles was calculated by averaging 50 nanoparticles selected on a TEM image using Nano Measurer software.

## Measurement of EMI shielding effectiveness

In order to test the EMI shielding effectiveness (EMI SE) in far-field radiation, we adopt vector network analyzer test method here. The EMI SE as well as the reflection (R), absorption (A) and transmission (T) coefficients were calculated from the S-parameters ($S_{11}$, $S_{21}$, $S_{12}$, $S_{22}$) measured by the network analyzer. The total EMI shielding effectiveness ($SE_T$) consists of reflection ($SE_R$), absorption ($SE_A$) and multiple reflections ($SE_{MR}$), where $SE_{MR}$ are counted as $SE_A$ when the total electromagnetic shielding effectiveness exceeds 15 dB.

For reflection, absorption and transmission coefficients:

$$R + T + A = 1 \tag{1}$$

$$R = ||S_{11}||^2 = ||S_{22}||^2 \tag{2}$$

$$T = ||S_{12}||^2 = ||S_{21}||^2 \tag{3}$$

The EMI shielding effectiveness (EMI $SE_T$), reflection ($SE_R$) and absorption ($SE_A$) can be calculated by the following equations:

$$SE_T = 10\lg\left(\frac{1}{T}\right) = 10\lg\left(\frac{1}{|S_{12}|^2}\right) \tag{4}$$

$$SE_R = 10\lg\left(\frac{1}{1-R}\right) = 10\lg\left(\frac{1}{1-|S_{11}|^2}\right) \tag{5}$$

$$SE_A = 10\lg\left(\frac{1-R}{T}\right) = 10\lg\left(\frac{1-|S_{11}|^2}{|S_{12}|^2}\right) \tag{6}$$

$$SE_T = SE_R + SE_A + SE_{MR} \tag{7}$$

## Finite element analysis

The temperature and stress distributions of the $LIG@NiFe_2O_4$ and $rGO/LIG@NiFe_2O_4$ composites were simulated by the software package COMSOL Multiphysics (version: 5.6). The $rGO/LIG@NiFe_2O_4$ and $LIG@NiFe_2O_4$ composites undergo laser-induced heating in this study. The incident heat flux from the laser is modeled as a spatially distributed heat source on the surface of these materials. The transient thermal response of the rGO is then analyzed, capturing key parameters at each calculation step. The maximum temperatures, along with the peak temperature difference across the rGO, are recorded during the heating process. Additionally, the temperature distribution and stress distribution across the entire rGO layers are stored at specified output time steps. The heat distribution of the laser beam is defined using a set of variables in a Gaussian distribution. The mathematical expression for this distribution is given by:

$$P = \frac{P_0(1 - R_c)}{\pi r_c^2} \exp\left(-\frac{(x - x_0)^2}{2r_c^2} - \frac{(y - y_0)^2}{2r_c^2}\right) \tag{8}$$

where $P_0$ is the total power of the laser beam, $R_c$ is reflection coefficient of graphene surface, and $r_c$ is the radius of laser spot. Under the assumption that the laser operates at a wavelength where the rGO is opaque, it implies that no light is transmitting through the material. As a consequence, the entire heat generated by the laser is deposited solely at the surface of the rGO. This consideration is significant, as it simplifies the heat transfer analysis by focusing on the absorption and retention of laser-generated heat within the material's surface layer. Throughout the simulation, the bottom of the substrate and the surrounding environment maintain a fixed temperature of 293.15 K. It is important to note that heat loss from the rGO occurs through radiation to the environment:

$$q = \gamma\beta(T^4 - T_{amb}^4) \tag{9}$$

where $\gamma$ is the surface emissivity, $\beta$ is the Stefan-Boltzmann constant (a predefined physical constant), and $T_{amb}$ is the ambient temperature. The surface emissivity of rGO is estimated to be around 0.025.

Based on the first law of thermodynamics, the heat transfer in substrate rGO layers is:

$$\rho c_p \frac{\partial T}{\partial t} + \rho c_p \mathbf{u} \cdot \nabla T + \nabla \cdot \mathbf{q} = Q \tag{10}$$

where $\rho$ is the density of rGO, $c_p$ is the heat capacity of rGO, the rate of energy flowing per unit area is:

$$\mathbf{q} = -k\nabla T \tag{11}$$

with $k$ being thermal conductivity, $Q$ is the internal heat energy per unit volume.

Due to the elevated temperature, the thermal expansion of rGO is described by:

$$\varepsilon_{th} = \alpha\left(T - T_{ref}\right) \tag{12}$$

where $\alpha$ is the thermal expansion coefficient. For rGO, the in-plane thermal expansion is negative, while the out-of-plane thermal expansion is positive (see Supplementary Table 3). Considering the thermal expansion, the stress of rGO layers is:

$$\sigma = E(\varepsilon - \varepsilon_{th}) \tag{13}$$

where $E$ is the elastic modulus, and $\varepsilon$ is the total strain.

The rGO is meshed using a free tetrahedral mesh. A finer mesh and tighter solver tolerances would give slightly more accurate predictions of the peak temperature. In the results visualization of the temperature and stress profile across the rGO, the results can be visualized in either the spatial frame or the material frame.

In addition, we assume that both NiFe$_2$O$_4$ particles and the substrate are rigid, thereby excluding thermal expansion effects. However, these components retain the ability to conduct and store thermal energy. All the detailed parameters are shown (see Supplementary Table 3).

## Data availability
The data generated in this study are available in the Source Data file. Source data are provided with this paper.

## Code availability
The code for the theoretical model is available via GitHub at https://github.com/edwinnewton/A-popcorn-making-mimic-strategy-for-graphene-NIFe2O4-films.

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

## Acknowledgements

The authors acknowledge the financial support from the National Natural Science Foundation of China (Project No. 52005289, 52273297 and 12302124), Guangdong Provincial Key Laboratory of Thermal Management Engineering & Materials (2020B1212060015), Shenzhen Natural Science Fund (20220809173605001), Shenzhen Science and Technology Program (GJHZ20210705143000002 & KJZD20230923115402005) and Shenzhen Geim Graphene Center for financial supports.

## Author contributions

Conceptualization: C.Y. and M.L.; Methodology: M.L., Z.W., Z.S., F.W., C.Y., Z.G., F.K.; Experiment: M.L., Z.W., F.W., Z.J., Z.S., H.Z., Z.G.; Funding acquisition: C.Y., Z.G.; Supervision: C.Y., Z.G.; Writing–original draft: M.L., F.W., Z.W., H.Z., C.Y.; Writing–review & editing: M.L., F.W., C.Y., Z.G., F.K.

## Competing interests

The authors declare no competing interests.
