## [Peer Review File · Nature Communications]

A “popcorn-making-mimic” strategy for compounding graphene@NiFe₂O₄ flexible films for strong electromagnetic interference shielding and absorptionREVIEWER COMMENTS

Reviewer #1 (Remarks to the Author):

In this manuscript, a highly conjugated and uniformly compounded graphene@NiFe₂O₄ composite film was fabricated by a laser-assisted method in ambient condition. This method can inhibit unwanted structural disintegration and mass loss, by avoiding oxidation, bursting, and inhomogeneous heat accumulations, thus can harvest a highly integrated composite structure with superior electrical conductivity and high saturated magnetization. It exhibited excellent EMI absorption capability and mechanical flexibility.

One of highlights in this manuscript is the composite fabrication method, that is, a laser-induced graphene film was used as a substrate, and then to make the graphene-loaded iron salt, nickel salt and GO composite film which was further processed by a fiber laser in ambient. The laser parameters were set as constant, i.e. a defocus of 10 mm, a scanning speed of 250 mm s⁻¹, a scanning pitch of 0.005 mm, and a power percentage of 6%. The reaction container was designed by introducing a GO lid on a pot, for the purpose of effective utilization of the laser pulse energy. During the reaction process of laser processing, the nearly insulating GO lid converts the local transient laser pulses energy into uniform thermal energy during the transition to rGO and confines it inside the LIG structure. This in turn allows the heat to uniformly diffuse inside LIG. Such as reaction space and constitution can effectively prevents violent combustion and uneven energy distribution, thus reducing the mass and energy loss during the drastic laser processing step.

Authors emphasize above composite preparation method and conditions, however, from present description and experimental results, it is hard to be a significant technique development in material preparation, many roles and effects are still unknown, such as the species role for protective lid, the size of reaction pot or local laser pulses energy, power range of laser beam, necessary conditions for nucleation and growth of nanoparticles, heat flow/diffusion and distribution, thermal limitations for volume expansion, combustion and bonding, etc. These issues should be well addressed, to raise the maturity and significance of the preparation method mentioned.

The mechanism for EMI performance of rGO/LIG@NiFe₂O₄ composites should be further discussed. Higher electrical conductivity causes a strong reflection of EW, magnetic/dielectric losses promote absorption of EM, in this multiscale composite, many factors can bring obvious EMI activities. At present, no rich evidence can confirm the rGO/LIG@NiFe₂O₄ composite is new "enough".

Reviewer #2 (Remarks to the Author):

In this work, the authors reported a two-step laser processing method to prepare rGO/LIG@NiFe₂O₄ composite film, i.e., the first laser processing step to prepare the LIG layer and the second laser processing step to prepare rGO/LIG@NiFe₂O₄ composite film on the GO-covered LIG layer deposited with NiFe salts. This strategy is interesting and effective, which can significantly suppress the oxidation/burning of the LIG layer during the second laser processing step and produce NiFe₂O₄ nanoparticles simultaneously. As-prepared rGO/LIG@NiFe₂O₄ show good electrical conductivity and high saturated magnetization, which exhibit impressive EMI shielding performance. In addition, the rGO/LIG@NiFe₂O₄ composite layer on the PI possesses high stability and flexibility, which is important for potential application. This work is noteworthy and may inspire studies in this area. The data can support the conclusions well. However, some issues need to be addressed, which are listed as follows.

1. During the fabrication of rGO/LIG@NiFe₂O₄, the NiCl₂ solution was first drop-casted on the LIG surface, and then the mixture of GO dispersion and Fe(NO₃)₃ solution was drop-casted on the dried NiCl₂-coated LIG. Some problems may exist in this process: 1) it seems the NiCl₂ is separated from Fe(NO₃)₃ due to the GO incorporation, how did they compound with each other? 2) the Fe(NO₃)₃ solution may cause the agglomeration of the GO, how did the authors solve this problem or does it influence the drop-casting process? 3) are the NiFe precursors distributed uniformly on the LIG surface? This may cause the nonuniform distribution of the NiFe₂O₄

nanoparticles.

2. Where are the NiFe₂O₄ nanoparticles distributed? In the manuscript, the authors described the NiFe₂O₄ nanoparticles were embedded in the LIG nanosheets, are there any NiFe₂O₄ nanoparticles embedded in the rGO layer? It is suggested to clarify.

3. How about the uniformity of the rGO/LIG@NiFe₂O₄ composite film? Did the samples produced at different region exhibit the same EMI shielding performance?

4. The detailed information should be provided at the first occurrence of the mark "F-N-x" in the manuscript.

Reviewer #3 (Remarks to the Author):

This manuscript presents a new concept of "popcorn-making-mimic" strategy to create an insulating graphene oxide (GO) "lid" for solving the problems of damaged conductive network of LIG and the non-uniform distribution of magnetic particles during the reaction process using laser treatment. This work also demonstrates the exceptional electromagnetic interference (EMI) shielding effectiveness of 51 dB with a film's thickness of 166 μm, which can find potential applications in electronics and communications industry. In addition, it is claimed that the "popcorn-making-mimic" strategy can be extended to other composite materials by adjusting the precursors and processing parameters, potential for versatile advanced applications. However, I am not sure if the mechanism of GO lid is properly related to the popcorn making process. Moreover, the manuscript is not well prepared with many ambiguities, especially regarding the essential role of GO lid and its related mechanisms. Therefore, I cannot recommend the publication of this manuscript in the current form. Some critical issues are listed below.

1. In the fabrication of composite films, nickel chloride hexahydrate aqueous solution was firstly dropped on the LIG, followed by drying in the oven. Then, the mixture solution of GO and ferric nitrate nonahydrate were drop-cast on the dried LIG film with nickel chloride hexahydrate, followed by drying in the oven. Afterwards, the above-mentioned dried composite film was laser treated to produce rGO/LIG@NiFe₂O₄ composite film. According to the schematic illustration in Fig. 1B and SEM image shown in Fig. 1G, LIG was covered by GO flakes. In this case, how to make sure an even distribution of ferric nitrate in LIG structure so that the ferric nitrate nonahydrate can mix with nickel chloride hexahydrate and react under laser process? In fact, the schematic in Fig. 1B did not illustrate the two step deposition of ferric nitrate nonahydrate and chloride hexahydrate. The authors should explain the necessity for the two step process and how to make sure the two can mix uniformly in the presence of GO flakes.

2. Page 5, line 165: The authors claimed "the as-obtained LIG maintains a rich three-dimensional (3D) structure with high crystallinity and higher conductivity (Fig. 1C, D)". However, the crystallinity cannot be identified through SEM images in Fig. 1C and D, nor the conductivity. The authors need to justify the claim with XRD and conductivity measurements.

3. For the SEM images shown in Fig. 1, the authors should clearly indicate if those images were taken from the surfaces or the cross sections.

4. Fig. 1E-H: The authors mentioned that NiFe₂O₄ nanoparticles could be seen from Fig. 1E-F. I assume those rough particles are the NiFe₂O₄. However, these particles were not clearly discernible from the SEM images in G and H. The authors need to clarify what those particles are in E and F, and point out where are the NiFe₂O₄ nanoparticles, if can be seen from SEM images, in G and H.

5. Fig. 1I and J: it is difficult to see any Ni in the EDX mapping. Moreover, what is the difference in terms of NiFe₂O₄ distribution on the surfaces of rGO and LIG?

6. Page 5, line 174-175: "Such a drastic difference can be explained by the accumulation of heat and bursting effect caused by the transient laser pulse." This sentence is confusing. What 'drastic difference' the authors were referring to? The context should be provided in a more clear manner.

7. Fig. 1M: The size distribution seems to be rather artificial because the counts for each bin are roughly the same.

8. Page 5, Line 156-157: The authors claimed that "..., the nearly insulating GO lid converts the local transient laser pulses energy into uniform thermal energy during the transition to rGO and confines it inside the LIG structure. This in turn allows the heat to uniformly diffuse inside LIG." Here, does 'insulating' mean 'thermally insulating'? If so, how does the thermal insulation of GO help in uniform heat distribution inside LIG? Details should be provided. It would be even better if the authors could show the temperature distribution using IR or other techniques to substantiate

the important role of GO lid in spatially uniform-distributed heat.

9. With the above in mind, the mechanism of heat retention is still not very clear. In the popcorn making process, the lid is to retain the heat generated in the pot by thermal insulation, but no heat is generated from the lid itself. In the current method, the GO "lid" also served as a photothermal converter, generating heat during the laser process. The thickness of GO layer was only a few nanometers, which may not be effective in reducing the heat loss. Therefore, the mechanism of popcorn making process might not fully apply in this situation. Evidence of heat retention and clear mechanisms should be provided.

10. Fig. 2C: The loading of NiFe₂O₄ seems to affect the reduction of GO instead of the graphitization of LIG nanosheets in the rGO/LIG@NiFe₂O₄ sample. A higher NiFe₂O₄ loading led to a lower degree of reduction as indicated by the increasing D/G peak intensity ratio. Please clarify if the increasing D/G ratio was due to the reduction of GO or graphitization of LIG nanosheets and explain why.

11. Page 7, Line 236 – 238: The authors stated that "as the concentration of the precursor salt increases, the loading of NiFe₂O₄ inside the composite film structure increases accordingly, leading to the corresponding increase in the thickness of the rGO/LIG@NiFe₂O₄ composite film (Supplementary Table 1)." However, the thicknesses of rGO/LIG@NiFe₂O₄ (F-N-0.3) (51 μm), rGO/LIG@NiFe₂O₄ (F-N-0.3) (67 μm), and rGO/LIG@NiFe₂O₄ (F-N-0.7) (70 μm) were smaller than that of LIG (71 μm). Please clarify.

12. The numbering of Supplementary figure and table was very random, which is not in sequence with their first appearance in the manuscript. In addition, some supplementary figures were not discussed at all in the main text, for instance Fig. S1, S3, S7, S8, S9, S12, S14, and Table S3.

13. Line 293: "...C1s and O1s of LIG and rGO/LIG@NiFe₂O₄ are also given (Supplementary Fig. 4)". It should be "Supplementary Fig. 5".

Response to Reviewers

We appreciate the detailed and constructive comments provided by the reviewers.

‘Please refer to the followings for all details.

Reviewer #1:

In this manuscript, a highly conjugated and uniformly compounded graphene@NiFe₂O₄ composite film was fabricated by a laser-assisted method in ambient condition. This method can inhibit unwanted structural disintegration and mass loss, by avoiding oxidation, bursting, and inhomogeneous heat accumulations, thus can harvest a highly integrated composite structure with superior electrical conductivity and high saturated magnetization. It exhibited excellent EMI absorption capability and mechanical flexibility. One of highlights in this manuscript is the composite fabrication method, that is, a laser-induced graphene film was used as a substrate, and then to make the graphene-loaded iron salt, nickel salt and GO composite film which was further processed by a fiber laser in ambient. The laser parameters were set as constant, i.e. a defocus of 10 mm, a scanning speed of 250 mm s⁻¹, a scanning pitch of 0.005 mm, and a power percentage of 6%. The reaction container was designed by introducing a GO lid on a pot, for the purpose of effective utilization of the laser pulse energy. During the reaction process of laser processing, the nearly insulating GO lid converts the local transient laser pulses energy into uniform thermal energy during the transition to rGO and confines it inside the LIG structure. This in turn allows the heat to uniformly diffuse inside LIG. Such as reaction space and constitution can effectively prevents violent combustion and uneven energy distribution, thus reducing the mass and energy loss during the drastic laser processing step.

Response: We thank the reviewer for the positive comments on this work.

Authors emphasize above composite preparation method and conditions, however, from present description and experimental results, it is hard to be a significant technique development in material preparation, many roles and effects are still unknown, such as the species role for protective lid, the size of reaction pot or local laser pulses energy, power range of laser beam, necessary conditions for nucleation and growth of nanoparticles, heat flow/diffusion and distribution, thermal limitations

for volume expansion, combustion and bonding, etc. These issues should be well addressed, to raise the maturity and significance of the preparation method mentioned.

Response: The reviewer raised questions regarding a series of laser processing parameters and the main innovation of this study. There have been many discussions regarding laser fabrication mechanisms and the parameters such as “*the size of reaction pot or local laser pulses energy, power range of laser beam*” can be set during the experiment, which had been optimized for the best performance that we could access.^{1,2} But, considering laser treatment is a transient process and at a high temperature, most available and powerful analytical methods (such as TEM, X-ray and calorimetry) can hardly be adopted for in-situ observation to reveal “*the species role for protective lid, necessary conditions for nucleation and growth of nanoparticles, heat flow/diffusion and distribution, thermal limitations for volume expansion, combustion and bonding, etc.*” To best address your question and more clearly elucidate the innovation of our study, we are glad to employ simulation method to semi-quantitatively analyze the related mechanisms involved during this “popcorn-making” process. Considering the time scale and dimensional characteristics, we established a container model as illustrated in the schematic diagram (Figure R1) using finite element modeling. Both rGO/LIG@NiFe₂O₄ (the sample with GO lid) and LIG@NiFe₂O₄ (the control sample without GO lid) undergo laser-induced heating in this study. Specifically, these composites undergo one second of heating from a 2.4 W laser, and the incident heat flux from the laser is modeled as a spatially distributed heat source on the surface of the material, which is given by:

$$P = \frac{P_0(1-R_c)}{\pi r_c^2} \exp\left(-\frac{(x-x_0)^2}{2r_c^2} - \frac{(y-y_0)^2}{2r_c^2}\right).$$

where P_0 is the total power of the laser beam, R_c is reflection coefficient of graphene surface, and r_c is the radius of laser spot.

Based on the first law of thermodynamics, the heat transfer in substrate rGO layers is:

$$\rho c_p \frac{\partial T}{\partial t} + \rho c_p \mathbf{u} \cdot \nabla T + \nabla \cdot \mathbf{q} = Q,$$

where ρ is the density of rGO, c_p is the heat capacity of rGO, the rate of energy flowing per unit area is:

$$\mathbf{q} = -k\nabla T$$

with k being the thermal conductivity, and Q is the internal heat energy per unit volume.

Due to the elevated temperature, the thermal expansion of rGO is described by:

$$\varepsilon_{th} = \alpha(T - T_{ref}),$$

where α is the thermal expansion coefficient. For rGO, the in-plane thermal expansion is negative, while the out-of-plane thermal expansion is positive. Considering the thermal expansion, the stress of rGO layers is:

$$\sigma = E(\varepsilon - \varepsilon_{th})$$

where E is the elastic modulus and ε is the total strain. The detailed parameters are shown in Table R1 (also in supplementary Table 3).

Figure. R1. (A-B) SEM images of rGO/LIG@NiFe₂O₄ (A: top-view, B: cross sectional-view). (C) The rGO/LIG@NiFe₂O₄ finite element model (The rGO/LIG@NiFe₂O₄ composite model being subjected to one second of heating from a 2.4 W laser)

Symbols	Meaning	Value	Reference
E	Elastic modulus of rGO	2.0 TPa	Ref ³
ν	Poisson's ratio of rGO	0.18	Ref ⁴
t	Thickness of rGO	1 μm	This article
ρ	Density of rGO	2.267 g/cm ³	Ref ⁵
σ_{th}	Strength of rGO	130 GPa	Ref ⁶

k	Thermal conductivity of rGO	$4000 \text{ W} \cdot \text{m}^{-1} \text{K}^{-1}$	Ref ⁷
γ	Emissivity of rGO	0.025	Ref ⁸
α	Coefficient of thermal expansion (in-plane) of rGO	$-2 \times 10^{-6} \text{ K}^{-1}$	Ref ⁶
α	Coefficient of thermal expansion (out-of-plane) of rGO	$6 \times 10^{-6} \text{ K}^{-1}$	Ref ⁶
R_c	Reflection coefficient of rGO	0.1	Ref ⁹
r_c	Radius of laser spot	$25 \text{ } \mu\text{m}$	This article
c_{p1}	Heat capacity of rGO	$1 \text{ J}/(\text{kg} \cdot \text{K})$	Ref ¹⁰
k_s	Thermal conductivity of substrate	$10 \text{ W} \cdot \text{m}^{-1} \text{K}^{-1}$	This article
c_{ps}	Heat capacity of substrate	$700 \text{ J}/(\text{kg} \cdot \text{K})$	This article
k_p	Thermal conductivity of NiFe ₂ O ₄	$50 \text{ W} \cdot \text{m}^{-1} \text{K}^{-1}$	This article
c_{p2}	Heat capacity of NiFe ₂ O ₄	$700 \text{ J}/(\text{kg} \cdot \text{K})$	This article

Table R1. Parameters for simulation

Using this model, the transient thermal response of the reduced graphene oxide (rGO) was analyzed, capturing key parameters at each calculation step. As a result, the temperature and stress distributions at the 1-second mark of the LIG@NiFe₂O₄ and rGO/LIG@NiFe₂O₄ composites are obtained. As shown in Fig. R2(a-b), it is evident that the rGO/LIG@NiFe₂O₄ composites exhibit considerably lower temperatures and stress levels in the rGO layers (excluding the lid) compared to the LIG@NiFe₂O₄ composite. This discrepancy can be attributed to the role of the rGO lid, which functions as a thermal insulator, effectively shielding and protecting the underlying rGO layers. The protective function of the rGO lid is further emphasized as it mitigates temperature and stress in the underneath rGO layers. To illustrate the temporal evolution of these parameters, Fig. R2(c-d) demonstrates the kinetics of the temperatures and maximum von Mises stress within the rGO layers. Notably, both temperature and stress reach a steady state within a remarkably short duration of 1 millisecond. Moreover, the rGO lid homogenizes the temperature of large-area underlying rGO layers.

Figure. R2. Simulation results of the the thermal induced failure in rGO. (a) The temperature and stress distribution in LIG@NiFe₂O₄ composite. (b) The temperature and stress distribution in rGO/LIG@NiFe₂O₄ composite. (c) The maximum temperature and (d) maximum vons Mises stress in rGO layers (excluded the lid) in LIG@NiFe₂O₄ composite and rGO/LIG@NiFe₂O₄ composite.

The pulsed power of laser triggers elevated temperature and stress levels during the heating process in the LIG@NiFe₂O₄ composite, as depicted in Fig. R3. In comparison, the temperature within the substrate rGO layer of the rGO/LIG@NiFe₂O₄ composite is notably lower. This discrepancy arises from the distinct thermodynamics in the two composites. In the rGO/LIG@NiFe₂O₄ composite, energy is stored in the rGO lid, which subsequently functions as a continuous heat source for the substrate rGO layer. Consequently, the rGO layer experiences a more controlled and moderated temperature increase in contrast to the LIG@NiFe₂O₄ composite. This observation underscores the significant role of the rGO lid in regulating and distributing thermal energy, contributing to a more controlled thermal response in the composite. Notably, the 28% v/v is estimated from our experimental data, and as a matter of fact, the NiFe₂O₄

particles have a very slight contribution of this thermodynamic process. Accordingly, we have added corresponding description about the simulation in section 2.5 of the main text.

Figure. R3. The effect of pulsed power on the failure of rGO. (a-b) The pulsed laser power with pulse width 2ns. (c) The maximum temperature and (d) maximum von Mises stress in rGO layers (excluded the lid) in LIG@NiFe₂O₄ composite, rGO/LIG@NiFe₂O₄ composite, and rGO/LIG.

Revision:

“2.5 GO lid mechanism analysis

To further elaborate on the role of the GO lid in regulating the heat distribution during laser processing, we developed a container model as illustrated in the schematic diagram (Fig. 4A) using finite element modeling. Specifically, the rGO/LIG@NiFe₂O₄ and LIG@NiFe₂O₄ composites undergo one second of heating from a 2.4 W laser. As shown in Fig. 4(B-C), the rGO/LIG@NiFe₂O₄ composites exhibit considerably lower temperatures and stress levels in the rGO layers (excluding the lid) compared to the LIG@NiFe₂O₄ composite. This discrepancy can be attributed to the role of the rGO lid,

which functions as a thermal insulator, effectively shielding and protecting the underlying rGO layers. The protective function of the rGO lid is further emphasized as it mitigates temperature and stress in the underneath rGO layers. To illustrate the temporal evolution of these parameters, Fig. 4(D-E) demonstrates the kinetics of the temperatures and maximum von Mises stress within the rGO layers. Significantly, both temperature and stress reach a steady state within a remarkably short duration of 1 millisecond. The thermal stress in the rGO layers, without the GO lid protection, approaches the theoretical strength limit (130 GPa, Supplementary Table 3). This condition promotes the development of defects, including flaws or microcracks. These defects can then coalesce to form continuous cracks that fragment the rGO layers, resulting in mass loss. Meanwhile, Fig. 4C also illustrates that the rGO lid homogenizes the temperature of the large-area underlying rGO layers.

In practical scenarios, the laser operates in a pulsed mode, with a power duration less than 1 millisecond. We performed simulations to analyze heat transfer and stress variation, taking into account the pulsed laser power with a duration of 2 nanoseconds. In the case of pulsed laser (Fig. 4F, G), the intermittent power of the laser triggers elevated temperature and stress levels during the heating process in the LIG@NiFe₂O₄ composite. In comparison, the temperature within the substrate rGO layer of the rGO/LIG@NiFe₂O₄ composite is notably lower (Fig. 4H). This discrepancy arises from the distinct thermodynamics in the two composites. In the rGO/LIG@NiFe₂O₄ composite, energy is stored in the rGO lid, which subsequently functions as a continuous heat source for the substrate rGO layer. Consequently, the rGO layer experiences a more controlled and moderated temperature increase in contrast to the LIG@NiFe₂O₄ composite. This observation underscores the significant role of the rGO lid in regulating and distributing thermal energy, contributing to a more controlled thermal response in the composite. Notably, the 28% v/v is estimated from our experimental data, and as a matter of fact, the NiFe₂O₄ particles have a very slight contribution of this thermodynamic process (Fig. 4I).”

To address the reviewer's comment regarding the significance of this technique

development in material preparation, this study demonstrates a novel approach to uniformly dispersing sub-10 nm nanoparticles within 3D porous matrices using transient lasers. This advancement addresses a complex challenge that is common to hydrothermal, CVD, and other synthesis methods, offering a potential breakthrough in the field. Our study introduces a GO lid to shift the energy input from a localized, pulsed manner to a sustained, widespread distribution. The GO lid also ensures uniform heat distribution and reduced peak temperatures during irradiation, as illustrated in Fig. R2. This method improves species retention by diffusing laser energy over a broader area, leading to a more consistent sub-10 nm nanoparticle size distribution, which is pivotal for applications demanding uniform nanomaterials. We have made revisions accordingly in the context so as to provide a more detailed description about the breakthroughs on materials preparations, which is listed below for your information.

Revision:

“In addition, the difference in thermal distribution between the two systems (the rGO/LIG@NiFe₂O₄ composite and the LIG@NiFe₂O₄ composite) directly affects the nucleation and growth process of nanoparticles. Generally, an increase in sintering temperature elevates the energy available to the particles, thereby enhancing the mobility of atoms or ions on their surfaces and leading to a higher diffusion rate. Consequently, higher temperatures accelerate the rate at which particles coalesce and grow, resulting in larger particle sizes within the composite. Regarding laser heating, the effect of laser power on particle size can be referred to the empirical formula: $D = K(P/v)^a$ (D is the crystallite dimension, K and a are constants related to the precursor salts, P is the laser power, and v is the laser scanning speed)^{11,12}. Without introducing the GO lid, the energy of the transient high-energy laser is injected into the system directly, resulting in an uneven heat distribution in the LIG. Consequently, the substantial temperature gradient causes pronounced disparities in particle size. Therefore, achieving a uniform formation of sub-10 nm nanoparticles within a 3D porous structure using transient lasers has been a formidable task, as it requires a stable and sufficiently low input power (P) coupled with an ultra-high laser scanning speed (v). In this study, the introduction of a GO lid addresses this issue by transforming

the energy input from a localized, intermittent process to a sustained, widespread one, akin to achieving a high laser scanning speed. This approach not only enhances the retention rate of the species by converting concentrated laser power into a more diffuse, stable input over a larger area but also results in the production of nanoparticles with a more uniform sub-10 nm size distribution. Therefore, our method provides a novel paradigm in the regulation of the microsystem environment of the reaction process and the preparation of nanomaterials, setting a new standard in the field.”

The mechanism for EMI performance of rGO/LIG@NiFe₂O₄ composites should be further discussed. Higher electrical conductivity causes a strong reflection of EW, magnetic/dielectric losses promote absorption of EM, in this multiscale composite, many factors can bring obvious EMI activities. At present, no rich evidence can confirm the rGO/LIG@NiFe₂O₄ composite is new “enough”.

Response: We appreciate the reviewer’s comment. The main breakthrough of this study is not to demonstrate a new EMI shielding mechanism. Instead, it highlights that through carefully engineering such an rGO/LIG@NiFe₂O₄ composite preparation method, we can enhance its normalized EMI absorption capability to a state-of-the-art level. This offers important insights for both the preparation of materials and the development of EMI shielding materials. Here we would like to discuss about the innovation of our sample preparation method to the EMI working mechanism.

The EM wave attenuation mechanisms of carbon-based composite materials mainly include conductivity loss, polarization loss, magnetic loss, and multiple scattering and reflection¹³. Considering that LIG is a conductive network with abundant surface defects and a three-dimensional porous structure, the multiple reflection effect for the shield with porous structures like the case of LIG cannot be ignored¹⁴. While the contribution of multiple reflections is included in absorption because the energy of multi-reflected radiation finally is absorbed or dissipated in the form of heat. In addition, rGO surfaces are rich in defects and surface functional groups; these functional groups and defects lead to asymmetric charge density differences, resulting in dipoles. Under an alternating electromagnetic field, these dipoles will absorb electromagnetic energy

and generate orientation rotation, which convert electromagnetic energy into thermal energy through relaxation, which is called “relaxation loss”^{15,16}. Maosheng Cao’s group has made much contribution to the design and mechanism understanding of electromagnetic shielding materials. For example, in a Cao’s work (2018, citations: 402)¹⁷, they said “*For an ideal microwave absorber, both highly efficient attenuation and well impedance matching are required, ..., the abundant interfaces among small Fe₃O₄ clusters and NG layers are considered as capacitor-like structure, and have great contribution to microwave attenuation*”. Similarly, the abundant heterostructures between NiFe₂O₄ nanoparticles, LIG, and rGO also can be considered as a capacitor-like structure, which improves impedance matching, reduces electromagnetic wave reflection, and contributes greatly to microwave attenuation¹⁸. In Cao’s another work (2018, citations: 416)¹⁹, they said “*Tailoring the NiFe₂O₄ clusters affects both the dielectric loss and magnetic loss, which directly determine the microwave attenuation performance, ..., According to the formula of micro-current network reported by Cao et al.²⁰, when the deposition ratio of magnetic clusters increases, the strengthened energy barrier for electron hopping hampers the formation of micro-current networks, resulting in poor conduction loss*”. So, both the amount and size of nanoparticles in the composite can substantially influence the electrical conductivity, which in turn change the electromagnetic shielding property. Recent related works published in high-impact journals also recognize the mechanism of electromagnetic shielding process above^{17,21-24}. Considering that our strategy here can well adjust the nanoparticle size and improve nanoparticle loading in the composite, it is a powerful method to obtain good impedance matching and microwave attenuation.

About your concern on the novelty of the rGO/LIG@NiFe₂O₄ composite, introducing a uniform distribution of sub-10 nm nanoparticles in such a porous graphene conductive network has long been a significant challenge; our “popcorn-making-mimic” strategy is particularly simple and efficient, which can render mild growth of nanoparticles by homogenizing the laser energy, reducing the air mass transfer, and restricting the formation of nanoparticles in a microreactor thus making them finer and more uniformly distributed. We also demonstrated that the materials obtained by this strategy

are able to significantly enhance the impedance matching while minimally disrupting the conductive channels. This would potentially become a versatile toolbox for developing novel nanocomposite materials in a green, controllable and efficient manner.

Reviewer #2:

In this work, the authors reported a two-step laser processing method to prepare rGO/LIG@NiFe₂O₄ composite film, i.e., the first laser processing step to prepare the LIG layer and the second laser processing step to prepare rGO/LIG@NiFe₂O₄ composite film on the GO-covered LIG layer deposited with NiFe salts. This strategy is interesting and effective, which can significantly suppress the oxidation/burning of the LIG layer during the second laser processing step and produce NiFe₂O₄ nanoparticles simultaneously. As-prepared rGO/LIG@NiFe₂O₄ show good electrical conductivity and high saturated magnetization, which exhibit impressive EMI shielding performance. In addition, the rGO/LIG@NiFe₂O₄ composite layer on the PI possesses high stability and flexibility, which is important for potential application. This work is noteworthy and may inspire studies in this area. The data can support the conclusions well. However, some issues need to be addressed, which are listed as follows.

Response: We appreciate the reviewer's positive comments. The questions raised by the reviewer are addressed point by point below.

1. During the fabrication of rGO/LIG@NiFe₂O₄, the NiCl₂ solution was first drop-casted on the LIG surface, and then the mixture of GO dispersion and Fe(NO₃)₃ solution was drop-casted on the dried NiCl₂-coated LIG. Some problems may exist in this process: 1) it seems the NiCl₂ is separated from Fe(NO₃)₃ due to the GO incorporation, how did they compound with each other? 2) the Fe(NO₃)₃ solution may cause the agglomeration of the GO, how did the authors solve this problem or does it influence the drop-casting process? 3) are the NiFe precursors distributed uniformly on the LIG surface? This may cause the non-uniform distribution of the NiFe₂O₄ nanoparticles.

Response: Thanks to the reviewer for such detailed questions. We are glad to address your concerns point by point.

1) Direct evidence that species can be homogeneously complexed with each other is provided from the optical image (Fig. R4), which shows that the ingredients are

uniformly complexed together.

2) Generally, GO is prepared in an acidic environment, thus it is quite stable in the $\text{Fe}(\text{NO}_3)_3$ solutions as well, which is also acidic. Besides, ultrasonic treatment before drop-coating can also help dispensing, which makes GO well capping on the surface of LIG.

3) LIG is very hydrophilic to both nickel chloride and ferric nitrate solution (Fig. R5), and thus the two solutions can be uniformly compounded inside its pores. Evidenced by the EDS mapping results of different regions, it can be seen that Ni and Fe are distributed uniformly. Therefore, we can deduce that the distribution of NiFe_2O_4 is uniform (Fig. R6).

Figure R4. Optical images of mixed solutions after drying in an oven at $60\text{ }^\circ\text{C}$. (A) $0.7\text{ M NiCl}_2 \cdot 6\text{H}_2\text{O} + 0.7\text{ M Fe}(\text{NO}_3)_3 \cdot 9\text{H}_2\text{O}$. (B) $0.7\text{ M NiCl}_2 \cdot 6\text{H}_2\text{O} + 0.7\text{ M Fe}(\text{NO}_3)_3 \cdot 9\text{H}_2\text{O} + 2\text{ mg/g GO}$.

Figure R5. Contact angle characterization results of different solutions on the LIG film. (a) 0.7 M $\text{NiCl}_2 \cdot 6\text{H}_2\text{O}$. (b) 0.7 M $\text{Fe}(\text{NO}_3)_3 \cdot 9\text{H}_2\text{O} + \text{GO}$

Figure R6. EDS mapping images of rGO/LIG@ NiFe_2O_4 composite film after stripping the rGO layer from different regions. (Blue for Fe, purple for Ni)

2. Where are the NiFe_2O_4 nanoparticles distributed? In the manuscript, the authors described the NiFe_2O_4 nanoparticles were embedded in the LIG nanosheets, are there any NiFe_2O_4 nanoparticles

embedded in the rGO layer? It is suggested to clarify.

Response: As the particle size of the NiFe₂O₄ nanoparticles is quite small (less than 10 nm), it is possible that some of the NiFe₂O₄ particles can be embedded in the rGO²⁵, as indicated in our mechanism diagram in Fig. 5C in the main text. But the rGO layer (~2 μm) is much thinner than the LIG layer (~70 μm), most of the NiFe₂O₄ nanoparticles are distributed in the LIG layer (Fig. R7).

Figure R7. TEM image showing the uniform distribution of NiFe₂O₄ on LIG of the rGO/LIG@NiFe₂O₄ composite film

3. How about the uniformity of the rGO/LIG@NiFe₂O₄ composite film? Did the samples produced at different region exhibit the same EMI shielding performance?

Response: 1) Based on the large number of SEM images, we did not see any significant agglomeration of NiFe₂O₄ particles. This is because that the wettability between the salts and the LIG surface is excellent, and thus the uniformity of rGO/LIG@NiFe₂O₄ composite film is largely dependent on the pore structure of LIG. To note, its pore distribution mainly depends on the power stability of laser processing, so a good control of laser power can ensure uniformity, which can be easily achieved.

2) Generally, the electromagnetic shielding test require the sample size as large as 20 mm x 20 mm; if the composite film had poor uniformity then the electromagnetic wave will not be able to gain effective magnetic loss, where the electromagnetic shielding performance will have less difference to the control sample. To note, the SE_T of LIG@NiFe₂O₄ is 27.0 dB, which is just 1.8 dB higher than pristine LIG, while the SE_T

of rGO/LIG@NiFe₂O₄ is 36.1 dB. If the nickel ferrate particles are not evenly distributed, it can severely degrade the electromagnetic shielding performance.

4. The detailed information should be provided at the first occurrence of the mark "F-N-x" in the manuscript.

Response: We thank the reviewer for the correction. We have revised the appropriate descriptions and scrutinized our manuscript.

Revision: *"As can be seen in Fig. 1K, the thickness of rGO/LIG@NiFe₂O₄ (F-N-0.7, "F" represents iron nitrate nonahydrate, "N" represents nickel chloride hexahydrate, and "X" represents the concentration of the composite films) is approximately 70 μm, which is almost the same as that of LIG (71 μm)."*

Reviewer #3:

This manuscript presents a new concept of "popcorn-making-mimic" strategy to create an insulating graphene oxide (GO) "lid" for solving the problems of damaged conductive network of LIG and the non-uniform distribution of magnetic particles during the reaction process using laser treatment. This work also demonstrates the exceptional electromagnetic interference (EMI) shielding effectiveness of 51 dB with a film's thickness of 166 μm, which can find potential applications in electronics and communications industry. In addition, it is claimed that the "popcorn-making-mimic" strategy can be extended to other composite materials by adjusting the precursors and processing parameters, potential for versatile advanced applications. However, I am not sure if the mechanism of GO lid is properly related to the popcorn making process. Moreover, the manuscript is not well prepared with many ambiguities, especially regarding the essential role of GO lid and its related mechanisms. Therefore, I cannot recommend the publication of this manuscript in the current form. Some critical issues are listed below.

Response: We appreciate the reviewer's comments on this work. As for your concern on the mechanism of GO lid, we established a container model as illustrated in the schematic diagram (Fig. R1) using finite element modeling to elucidate the difference of heat and stress distributions between rGO/LIG@NiFe₂O₄ and LIG@NiFe₂O₄, so that

to ensure that our proposed mechanism is widely acceptable. Specifically, the rGO/LIG@NiFe₂O₄ and LIG@NiFe₂O₄ composites undergo one second of heating from a 2.4 W laser, and the incident heat flux from the laser is modeled as a spatially distributed heat source on the surface of the material, which is given by:

$$P = \frac{P_0(1-R_c)}{\pi r_c^2} \exp\left(-\frac{(x-x_0)^2}{2r_c^2} - \frac{(y-y_0)^2}{2r_c^2}\right).$$

where P_0 is the total power of the laser beam, R_c is reflection coefficient of graphene surface, and r_c is the radius of laser spot.

Based on the first law of thermodynamics, the heat transfer in substrate rGO layers is

$$\rho c_p \frac{\partial T}{\partial t} + \rho c_p \mathbf{u} \cdot \nabla T + \nabla \cdot \mathbf{q} = Q,$$

where ρ is the density of rGO, c_p is the heat capacity of rGO, the rate of energy flowing per unit area is

$$\mathbf{q} = -k\nabla T$$

with k being thermal conductivity, and Q is the internal heat energy per unit volume. Due to the elevated temperature, the thermal expansion of rGO is described by

$$\varepsilon_{th} = \alpha(T - T_{ref}),$$

where α is the thermal expansion coefficient. For rGO, the in-plane thermal expansion is negative, while the out-of-plane thermal expansion is positive. Considering the thermal expansion, the stress of rGO layers is

$$\sigma = E(\varepsilon - \varepsilon_{th})$$

where E is the elastic modulus, and ε is the total strain. The detailed parameters are shown in Table R1.

Utilizing this model, we analyzed the internal distribution of LIG density/volume and characteristic dimensions during the laser fabrication process in the atmosphere. As can be seen in the Fig. R2, in the case of laser with constant input power, the rGO/LIG@NiFe₂O₄ composites exhibit considerably lower temperatures and stress levels in the rGO layers (excluding the lid) compared to the LIG@NiFe₂O₄ composite. While in the case of laser with pulsed power (Fig. R3), the protective function of the rGO lid is further emphasized as it mitigates temperature and stress in the underneath rGO layers. The laser energy is stored in the rGO lid, which subsequently functions as

a continuous heat source for the substrate rGO layer. Consequently, the rGO layer experiences a more controlled and moderated temperature increase in contrast to the LIG@NiFe₂O₄ composite. This observation underscored the significant role of the rGO lid in regulating and distributing thermal energy, contributing to a more controlled thermal response in the composite. And we have added corresponding description about the simulation in section 2.5 of the main text.

Revision:

“2.5 GO lid mechanism analysis

To further elaborate on the role of the GO lid in regulating the heat distribution during laser processing, we developed a container model as illustrated in the schematic diagram (Fig. 4A) using finite element modeling. Specifically, the rGO/LIG@NiFe₂O₄ and LIG@NiFe₂O₄ composites undergo one second of heating from a 2.4 W laser. As shown in Fig. 4(B-C), the rGO/LIG@NiFe₂O₄ composites exhibit considerably lower temperatures and stress levels in the rGO layers (excluding the lid) compared to the LIG@NiFe₂O₄ composite. This discrepancy can be attributed to the role of the rGO lid, which functions as a thermal insulator, effectively shielding and protecting the underlying rGO layers. The protective function of the rGO lid is further emphasized as it mitigates temperature and stress in the underneath rGO layers. To illustrate the temporal evolution of these parameters, Fig. 4(D-E) demonstrates the kinetics of the temperatures and maximum von Mises stress within the rGO layers. Significantly, both temperature and stress reach a steady state within a remarkably short duration of 1 millisecond. The thermal stress in the rGO layers, without the GO lid protection, approaches the theoretical strength limit (130GPa, Supplementary Table 3). This condition promotes the development of defects, including flaws or microcracks. These defects can then coalesce to form continuous cracks that fragment the rGO layers, resulting in mass loss. Meanwhile, Fig. 4C also illustrates that the rGO lid homogenizes the temperature of large-area underlying rGO layers.

In practical scenarios, the laser operates in a pulsed mode, with a power duration less than 1 millisecond. We performed simulations to analyze heat transfer and stress variation, taking into account the pulsed laser power with a duration of 2 nanoseconds.

In the case of pulsed laser (Fig. 4F, G), the intermittent power of the laser triggers elevated temperature and stress levels during the heating process in the LIG@NiFe₂O₄ composite. In comparison, the temperature within the substrate rGO layer of the rGO/LIG@NiFe₂O₄ composite is notably lower (Fig. 4H). This discrepancy arises from the distinct thermodynamics in the two composites. In the rGO/LIG@NiFe₂O₄ composite, energy is stored in the rGO lid, which subsequently functions as a continuous heat source for the substrate rGO layer. Consequently, the rGO layer experiences a more controlled and moderated temperature increase in contrast to the LIG@NiFe₂O₄ composite. This observation underscores the significant role of the rGO lid in regulating and distributing thermal energy, contributing to a more controlled thermal response in the composite. Notably, the 28% v/v is estimated from our experimental data, and as a matter of fact, the NiFe₂O₄ particles have a very slight contribution of this thermodynamic process (Fig. 4I).

In addition, the difference in thermal distribution between the two systems (the rGO/LIG@NiFe₂O₄ composite and the LIG@NiFe₂O₄ composite) directly affects the nucleation and growth process of nanoparticles. Generally, an increase in sintering temperature elevates the energy available to the particles, thereby enhancing the mobility of atoms or ions on their surfaces and leading to a higher diffusion rate. Consequently, higher temperatures accelerate the rate at which particles coalesce and grow, resulting in larger particle sizes within the composite. Regarding laser heating, the effect of laser power on particle size can be referred to the empirical formula: $D = K(P/v)^a$ (D is the crystallite dimension, K and a are constants related to the precursor salts, P is the laser power, and v is the laser scanning speed)^{11,12}. Without introducing the GO lid, the energy of the transient high-energy laser is injected into the system directly, resulting in an uneven heat distribution in the LIG. Consequently, the substantial temperature gradient causes pronounced disparities in particle size. Therefore, achieving a uniform formation of sub-10 nm nanoparticles within a 3D porous structure using transient lasers has been a formidable task, as it requires a stable and sufficiently low input power (P) coupled with an ultra-high laser scanning speed (v). In this study, the introduction of a GO lid addresses this issue by transforming

the energy input from a localized, intermittent process to a sustained, widespread one, akin to achieving a high laser scanning speed. This approach not only enhances the retention rate of the species by converting concentrated laser power into a more diffuse, stable input over a larger area but also results in the production of nanoparticles with a more uniform sub-10 nm size distribution. Therefore, our method provides a novel paradigm in the regulation of the microsystem environment of the reaction process and the preparation of nanomaterials, setting a new standard in the field.”

As for your remark: “*The manuscript is not well prepared with many ambiguities, especially regarding the essential role of GO lid and its related mechanism*”, we have deliberately revised the context to ensure that reviewers’ concerns. Please refer to the manuscript for all revision details.

1. In the fabrication of composite films, nickel chloride hexahydrate aqueous solution was firstly dropped on the LIG, followed by drying in the oven. Then, the mixture solution of GO and ferric nitrate nonahydrate were drop-cast on the dried LIG film with nickel chloride hexahydrate, followed by drying in the oven. Afterwards, the above-mentioned dried composite film was laser treated to produce rGO/LIG@NiFe₂O₄ composite film. According to the schematic illustration in Fig. 1B and SEM image shown in Fig. 1G, LIG was covered by GO flakes. In this case, how to make sure an even distribution of ferric nitrate in LIG structure so that the ferric nitrate nonahydrate can mix with nickel chloride hexahydrate and react under laser process?

In fact, the schematic in Fig. 1B did not illustrate the two step deposition of ferric nitrate nonahydrate and chloride hexahydrate. The authors should explain the necessity for the two step process and how to make sure the two can mix uniformly in the presence of GO flakes.

Response: This is a good question. Based on our contact angle test result (Fig. R5), LIG is very hydrophilic to both nickel chloride hexahydrate and ferric nitrate nonahydrate salt solutions, it can absorb two salt solutions through capillary force quickly. However, to eliminate any hollowing inside the LIG during the absorbing step, we split the drop coating step into two, where the first step only involve just one salt specie to be absorbed. This would make us more easily observe the absorption homogeneity (whether there is any hollowing) by optical microscope before dropping

GO (dark in color) on top.

So actually, the drop-coating process does not necessarily have to be a two-step one; and if there is no any hollowing concern, one drop-coating step would be even a better choice. Nevertheless, the NiCl_2 salt that is dried in the oven in the first step will be redissolved in the subsequent addition of a mixture of $\text{Fe}(\text{NO}_3)_3$ and GO flakes, compounding excellently with $\text{Fe}(\text{NO}_3)_3$, as can be seen from Fig. R4. We have made revisions in the manuscript accordingly to avoid any misleads.

Revision: *“Then, A mixture of 300 μL GO and ferric nitrate nonahydrate were drop-casted on the dried graphene film containing nickel chloride hexahydrate, and then placed in an oven at 60°C for 1 h. Notably, a simple step of drop-coating all three components can also be employed which can lead to the same result; yet splitting into two steps can facilitate optical microscopic observations to eliminate hollowing in the LIG area after the first step. The above-mentioned dried graphene-loaded iron salt, nickel salt and GO composite film was placed in a fiber laser processing platform and processed in ambient.”*

2. Page 5, line 165: The authors claimed “the as-obtained LIG maintains a rich three-dimensional (3D) structure with high crystallinity and higher conductivity (Fig. 1C, D)”. However, the crystallinity cannot be identified through SEM images in Fig. 1C and D, nor the conductivity. The authors need to justify the claim with XRD and conductivity measurements.

Response: Thanks for the reviewer’s comment. The evidence demonstrating high crystallinity and conductivity can be found from Supplementary Fig. 9 and Supplementary Fig. 10. To provide further evidence, we also included Fig. R8 below for your reference. Meanwhile, we have revised the description in the manuscript accordingly.

Revision: *“the as-obtained LIG maintains a rich three-dimensional (3D) structure (Fig. 1C, D) with higher conductivity and crystallinity (Supplementary Fig. 9, 10)”.*

Figure R8. Crystallinity and conductivity characterizations. (A) XRD pattern of LIG@NiFe₂O₄ and LIG. (B) Conductivity and resistance data of LIG@NiFe₂O₄ and LIG.

3. For the SEM images shown in Fig. 1, the authors should clearly indicate if those images were taken from the surfaces or the cross sections.

Response: We appreciate the reviewer's comment. We have added corresponding figures' description in the revised manuscript, and we also include the revised figures below for your reference (Fig. R9).

Figure R9. (A-B) SEM images of LIG with different magnifications (top-view). (C-D) SEM overview images of LIG@NiFe₂O₄ with different magnifications (top-view). (E) SEM overview images of rGO/LIG@NiFe₂O₄. (F) High magnification SEM image of internal LIG (top-view).

4. Fig. 1E-H: The authors mentioned that NiFe₂O₄ nanoparticles could be seen from Fig. 1E-F. I assume those rough particles are the NiFe₂O₄. However, these particles were not clearly discernible from the SEM images in G and H. The authors need to clarify what those particles are in E and F, and point out where are the NiFe₂O₄ nanoparticles, if can be seen from SEM images, in G and H.

Response: Thanks for the comments. SEM may not have high enough resolution to demonstrate the existence of NiFe_2O_4 particles in the $\text{rGO/LIG@NiFe}_2\text{O}_4$ samples, as these particles only have a diameter of about 2-5 nm (Fig. 1M). But the EDS images of the sample can clearly indicate that both Ni and Fe elements are very well distributed (Fig. R10).

Figure R10. SEM-EDS mapping images of $\text{rGO/LIG@NiFe}_2\text{O}_4$ composite film being stripped from the rGO layer.

5. Fig. 1I and J: it is difficult to see any Ni in the EDX mapping. Moreover, what is the difference in terms of NiFe_2O_4 distribution on the surfaces of rGO and LIG?

Response:

1) The color referring to the Ni element was difficult to see is due to the pixel loss when converting the word file into PDF. As can be seen in the following high-resolution EDX mapping image, there is indeed a uniform distribution of Ni elements on the surface and inside the $\text{rGO/LIG@NiFe}_2\text{O}_4$ composite film (Fig. R11). We have also added this image in the supporting information (Supplementary Fig. 17).

2) Since the layer of rGO is relatively thin, it is understandable that the NiFe_2O_4 particles it can load are quite limited and most of the NiFe_2O_4 nanoparticles will be distributed in the LIG (see Fig. 1O in the main text).

Figure R11. SEM images and corresponding (Ni) EDX mapping results of rGO and LIG. (A) Top-view from the rGO lid surface. (B) Cross sectional-view of LIG.

6. Page 5, line 174-175: “Such a drastic difference can be explained by the accumulation of heat and bursting effect caused by the transient laser pulse.” This sentence is confusing. What ‘drastic difference’ the authors were referring to? The context should be provided in a more clear manner.

Response: Thanks for your comment. The “drastic difference” refers to the difference in morphology between LIG@NiFe₂O₄ and rGO/LIG@NiFe₂O₄. We have carefully revised our manuscript to better improve the logic in the context. The following context is for your reference.

Revision: “the drastic morphological differences between rGO/LIG@NiFe₂O₄ and LIG@NiFe₂O₄ can be ascribed to the accumulation of heat and bursting effect caused by the laser pulse.”

7. Fig. 1M: The size distribution seems to be rather artificial because the counts for each bin are roughly the same.

Response: The data in Fig. 1M were counted manually and here are the corresponding original sizes of nanoparticle for your information (Table R2). Meanwhile, we provided the original high-resolution TEM images of different regions, as shown in the SEM images, their size distribution is consistent (Fig. R12).

No.	Particle size/nm
1	5.08
2	3.75
3	4.30
4	4.65
5	4.46
6	3.21
7	4.06
8	2.17
9	3.32
10	2.32
11	5.17
12	2.44
13	2.74
14	5.09
15	5.38
16	2.62
17	3.50
18	3.76
19	7.01
20	3.69
21	5.32
22	4.79
23	4.04
24	2.84
25	3.29
26	4.05
27	2.90

28	3.34
29	3.75
30	2.66
31	2.09
32	5.38
33	3.57
34	5.18
35	4.77
36	2.66
37	2.38
38	4.71
39	3.75
40	4.12
41	2.07
42	3.94
43	2.42
44	2.42
45	3.18
46	2.77
47	3.11
48	3.73
49	3.79
50	4.26
51	2.53

Table R2. The size of nickel ferrate nanoparticles

Figure R12. TEM images of rGO/LIG@NiFe₂O₄ composite film at different magnifications. (Fig. R12B is a zoom-in figure of the red rectangle zone of Fig. R12A)

8. Page 5, Line 156-157: The authors claimed that “..., the nearly insulating GO lid converts the local transient laser pulses energy into uniform thermal energy during the transition to rGO and confines it inside the LIG structure. This in turn allows the heat to uniformly diffuse inside LIG.” Here, does ‘insulating’ mean ‘thermally insulating’? If so, how does the thermal insulation of GO help in uniform heat distribution inside LIG? Details should be provided. It would be even better if the authors could show the temperature distribution using IR or other techniques to substantiate the important role of GO lid in spatially uniform-distributed heat.

Response: Thanks for the valuable comments. We used the term 'insulating' to indicate that the GO lid serves as a barrier, inhibiting heat flow between the LIG's interstitial space and the surrounding environment. This is due to the low thermal conductivity of the out-of-plane GO surface and its extensive surface coverage. As indicated in the simulation results (Fig. R2), the rGO lid actually functioned as a thermal insulator, effectively shielding and protecting the underlying rGO layers. The protective function of the rGO lid is further emphasized as it mitigates temperature and stress in the underneath rGO layers. We have added corresponding description about the simulation to illustrate the role of the GO lid in section 2.5 in the main text. For details, without introducing the GO lid, the energy of the transient high-energy laser is injected into the system directly, resulting in the inhomogeneous distribution of heat in the LIG (as shown in Fig. 4B), and thus the high temperature gradient leads to a large difference in

the size of the particles. Achieving a uniform formation of sub-10 nm nanoparticles within a 3D porous structure using transient lasers has traditionally been a formidable task. This is also a common challenge shared by e.g. hydrothermal reaction, CVD and other preparation methods. In this study, the introduction of a GO lid addresses this issue by transforming the energy input from a localized, intermittent process to a sustained, widespread one, akin to achieving a high laser scanning speed. Concurrently, the GO lid ensures a homogenized heat distribution within the LIG area and lowers the peak temperature during laser irradiation (as shown in Fig. 4C). This approach not only enhances the retention rate of the species by converting concentrated laser power into a more diffuse, stable input over a larger area but also results in the production of nanoparticles with a more uniform sub-10 nm size distribution. Therefore, our method provides a new paradigm in the regulation of the microsystem environment of the reaction process and the preparation of nanomaterials, setting a new standard in the field. Please refer to the revised manuscript for all revision details.

9. With the above in mind, the mechanism of heat retention is still not very clear. In the popcorn making process, the lid is to retain the heat generated in the pot by thermal insulation, but no heat is generated from the lid itself. In the current method, the GO "lid" also served as a photothermal converter, generating heat during the laser process.

The thickness of GO layer was only a few nanometers, which may not be effective in reducing the heat loss. Therefore, the mechanism of popcorn making process might not fully apply in this situation.

Evidence of heat retention and clear mechanisms should be provided.

Response: We can understand the reviewer's concern regarding the mechanistic explanation. To address reviewer's concern, we established a container model as illustrated in the schematic diagram (Fig. R1) with an aim to elucidate the role of GO lid during laser processing. Although the actual GO lid is about 2 microns thick (Supplementary Table 2.), as can be seen in the Fig. R2, it is evident that the rGO/LIG@NiFe₂O₄ composites exhibit considerably lower temperatures and stress levels in the rGO layers (excluding the lid) compared to the LIG@NiFe₂O₄ composite, which exactly demonstrated that the GO lid have a certain effect on homogenizing the

heat and stress distributions in the interstitial part of LIG. In addition, in the case of laser with pulsed power, for the rGO/LIG@NiFe₂O₄ composite, energy is stored in the rGO lid, which subsequently functions as a continuous heat source for the substrate rGO layer. Consequently, the rGO layer experiences a more controlled and moderated temperature increase in contrast to the LIG@NiFe₂O₄ composite (Fig. R3). All the results underscores the significant role of the rGO lid in regulating and distributing thermal energy, contributing to a more controlled thermal response in the composite.

10. Fig. 2C: The loading of NiFe₂O₄ seems to affect the reduction of GO instead of the graphitization of LIG nanosheets in the rGO/LIG@NiFe₂O₄ sample. A higher NiFe₂O₄ loading led to a lower degree of reduction as indicated by the increasing D/G peak intensity ratio. Please clarify if the increasing D/G ratio was due to the reduction of GO or graphitization of LIG nanosheets and explain why.

Response: To begin with, Raman is a surface examination method with a detection depth of typically only about 10 nm. Therefore, the growth of I_D/I_G with the increasing concentrations of Ni and Fe salts is mainly attributed to the effect of the reduction of the surface GO layer. As can be seen in the figure below (Fig. R13), the G band of the rGO/LIG@NiFe₂O₄ sample is red-shifted (from 1598 cm⁻¹ to 1583 cm⁻¹) compared to the pristine GO. Usually, the G-band shift of carbon-based composites can be attributed to the electron transfer of carbon materials and other compounds²⁶, so the significant Raman shift of rGO after anchoring with NiFe₂O₄ nanoparticles indicates the electrical interaction between nickel ferrate nanoparticles and rGO sheets²⁷. As can be seen from Fig. 2B in the main text, the intensity of the D peak gradually increases with the increase of precursor salt concentration, the D peak originates from the defects, edges, or the A_{1g}-symmetric vibration mode of the disordered graphite²⁸. So, the I_D/I_G intensity ratio grows from 0.84 (F-N-0.3) to 1.19 (F-N-1) mainly attributed to the NiFe₂O₄ nanoparticles' stresses on the surface of the rGO and the inducement of more defects and disorders²⁹.

Figure R13. Raman spectra around the G band area of GO and F-N-0.3 samples

11. Page 7, Line 236 – 238: The authors stated that “ as the concentration of the precursor salt increases, the loading of NiFe_2O_4 inside the composite film structure increases accordingly, leading to the corresponding increase in the thickness of the $\text{rGO/LIG@NiFe}_2\text{O}_4$ composite film (Supplementary Table 1).” However, the thicknesses of $\text{rGO/LIG@NiFe}_2\text{O}_4$ (F-N-0.3) ($51\ \mu\text{m}$), $\text{rGO/LIG@NiFe}_2\text{O}_4$ (F-N-0.5) ($67\ \mu\text{m}$), and $\text{rGO/LIG@NiFe}_2\text{O}_4$ (F-N-0.7) ($70\ \mu\text{m}$) were smaller than that of LIG ($71\ \mu\text{m}$). Please clarify.

Response: Thanks for the comment. Our statement in the manuscript that the corresponding $\text{rGO/LIG@NiFe}_2\text{O}_4$ composite thickness increases as the concentration of the precursor salt increases refers to composite film relative to low concentrations of precursor salts (Fig. R14). The fact is that they are all smaller than the pristine LIG film due to the partial structural collapse during the secondary processing.

Figure R14. Cross-sectional SEM images of different $\text{rGO/LIG@NiFe}_2\text{O}_4$ samples

12. The numbering of Supplementary figure and table was very random, which is not in sequence

with their first appearance in the manuscript. In addition, some supplementary figures were not discussed at all in the main text, for instance Fig. S1, S3, S7, S8, S9, S12, S14, and Table S3.

Response: We thank the reviewer's efforts in pointing out this discrepancy. We have revised the corresponding descriptions and scrutinized our manuscript.

Revision:

1) *"The rGO/LIG@NiFe₂O₄ composite film is slim and flexible, allowing for reshaping or bending into desirable structures (Supplementary Fig. 1)."*

2) *"The composite film was processed under the laser parameters with a defocus of 10 mm, a scanning speed of 250 mm s⁻¹, a scanning pitch of 0.005 mm, and a power percentage of 6% (The power was 2.43 W) (Supplementary Fig. 3 shows the exploration of suitable processing parameters)."*

3) *"...rGO/LIG@NiFe₂O₄ achieves an ultra-high EMI shielding effectiveness of 51 dB and 73% absorption at a thickness of 166 μm (Supplementary Fig. 8)"*

4) *"As can be seen, the as-obtained LIG maintains a rich three-dimensional (3D) structure (Fig. 1C, D) which provide information regarding higher conductivity and high crystallinity characteristics (Supplementary Fig. 9, 10)."*

5) *"LIG has excellent hydrophilicity, and its water contact angle is only 17°, which guarantees the sufficient infiltration of the precursor salt solution. After loading NiFe₂O₄ nanoparticles, the water contact angles of LIG@NiFe₂O₄ and rGO/LIG@NiFe₂O₄ composite films change to 20.7° and 53.5°, respectively (Supplementary Fig. 12). This shows that the loading of NiFe₂O₄ nanoparticles improves the hydrophobicity. In addition, due to the thermal reduction of oxygen-containing functional groups after laser treatment, the surface of rGO/LIG@NiFe₂O₄ composite films are more hydrophobic compared to LIG@NiFe₂O₄. For the electromagnetic shielding materials to be applied to devices and electronic devices in the future, the better hydrophobicity is favorable for the protection of the devices.*

"

6) *"After 10,000 bending cycles with a bending diameter of approximately 2 cm (Supplementary Fig. 14), the changes in square resistance for LIG and rGO/LIG@NiFe₂O₄ (F-N-0.7) samples were approximately 16.4% and 2.6%*

respectively.”

7) “In order to further demonstrate its advantages regarding the future critical performance indexes for EMI shielding uses, a six-dimensional radar plot is presented here (Fig. 5B, detailed data in Supplementary Table 4).”

13. Line 293: “...C1s and O1s of LIG and rGO/LIG@NiFe₂O₄ are also given (Supplementary Fig. 4)”. It should be “Supplementary Fig. 5”.

Response: We have revised the corresponding descriptions in the manuscript.

Revision: “High-resolution XPS spectra of C1s and O1s of LIG and rGO/LIG@NiFe₂O₄ are also given (Supplementary Fig. 5).”

Reference

- 1 Le, T.-S. D. *et al.* Recent Advances in Laser-Induced Graphene: Mechanism, Fabrication, Properties, and Applications in Flexible Electronics. *Advanced Functional Materials* **32**, 2205158, doi:<https://doi.org/10.1002/adfm.202205158> (2022).
- 2 Trusovas, R. *et al.* Recent Advances in Laser Utilization in the Chemical Modification of Graphene Oxide and Its Applications. *Advanced Optical Materials* **4**, 37-65, doi:<https://doi.org/10.1002/adom.201500469> (2016).
- 3 Lee, J.-U., Yoon, D. & Cheong, H. Estimation of Young's Modulus of Graphene by Raman Spectroscopy. *Nano Lett.* **12**, 4444-4448, doi:10.1021/nl301073q (2012).
- 4 Cao, G. Atomistic Studies of Mechanical Properties of Graphene. *Polymers* **6**, 2404-2432 (2014).
- 5 Torrisi, L., Cutroneo, M., Torrisi, A. & Silipigni, L. Measurements on Five Characterizing Properties of Graphene Oxide and Reduced Graphene Oxide Foils. *physica status solidi (a)* **219**, 2100628, doi:<https://doi.org/10.1002/pssa.202100628> (2022).
- 6 Chen, H., Mi, G., Li, P., Huang, X. & Cao, C. Microstructure and Tensile Properties of Graphene-Oxide-Reinforced High-Temperature Titanium-Alloy-Matrix Composites. *Materials* **13**, 3358 (2020).
- 7 Sang, M., Shin, J., Kim, K. & Yu, K. J. Electronic and Thermal Properties of Graphene and Recent Advances in Graphene Based Electronics Applications. *Nanomaterials* **9**, 374 (2019).
- 8 Krishna, A. *et al.* Ultraviolet to Mid-Infrared Emissivity Control by Mechanically Reconfigurable Graphene. *Nano Lett.* **19**, 5086-5092, doi:10.1021/acs.nanolett.9b01358 (2019).
- 9 Zeranska-Chudek, K. *et al.* Study of the absorption coefficient of graphene-polymer composites. *Scientific Reports* **8**, 9132, doi:10.1038/s41598-018-27317-0 (2018).
- 10 Elsaid, K. *et al.* Thermophysical properties of graphene-based nanofluids. *International Journal of Thermofluids* **10**, 100073, doi:<https://doi.org/10.1016/j.ijft.2021.100073> (2021).
- 11 Rodica, A., Ion, G. M., Popescu, C. R. & Ion, N. V. in *Proc. SPIE.* 174-181.
- 12 Alexandrescu, R. *et al.* Cu - Ni oxides obtained by laser and thermal processing of mixed salts. *Journal of Physics D: Applied Physics* **30**, 2620, doi:10.1088/0022-3727/30/18/017 (1997).
- 13 Xia, Y., Gao, W. & Gao, C. A Review on Graphene-Based Electromagnetic Functional Materials: Electromagnetic Wave Shielding and Absorption. *Advanced Functional Materials* **32**, 2204591, doi:<https://doi.org/10.1002/adfm.202204591> (2022).
- 14 Yu, W., Peng, Y., Cao, L., Zhao, W. & Liu, X. Free-standing laser-induced graphene films for high-performance electromagnetic interference shielding. *Carbon* **183**, 600-611, doi:<https://doi.org/10.1016/j.carbon.2021.07.055> (2021).
- 15 Cao, M.-S. *et al.* Electronic Structure and Electromagnetic Properties for 2D Electromagnetic Functional Materials in Gigahertz Frequency. *Annalen der*

- Physik* **531**, 1800390, doi:<https://doi.org/10.1002/andp.201800390> (2019).
- 16 Cao, W.-Q., Wang, X.-X., Yuan, J., Wang, W.-Z. & Cao, M.-S. Temperature dependent microwave absorption of ultrathin graphene composites. *Journal of Materials Chemistry C* **3**, 10017–10022, doi:10.1039/C5TC02185E (2015).
- 17 Wang, X.-X., Ma, T., Shu, J.-C. & Cao, M.-S. Confinedly tailoring Fe₃O₄ clusters-NG to tune electromagnetic parameters and microwave absorption with broadened bandwidth. *Chemical Engineering Journal* **332**, 321–330, doi:<https://doi.org/10.1016/j.cej.2017.09.101> (2018).
- 18 Shi, X.-L. *et al.* Nonlinear resonant and high dielectric loss behavior of CdS / α -Fe₂O₃ heterostructure nanocomposites. *Applied Physics Letters* **93**, 183118, doi:10.1063/1.3023074 (2008).
- 19 Zhang, Y., Wang, X. & Cao, M. Confinedly implanted NiFe₂O₄-rGO: Cluster tailoring and highly tunable electromagnetic properties for selective-frequency microwave absorption. *Nano Research* **11**, 1426–1436, doi:10.1007/s12274-017-1758-1 (2018).
- 20 Wen, B. *et al.* Temperature dependent microwave attenuation behavior for carbon-nanotube/silica composites. *Carbon* **65**, 124–139, doi:<https://doi.org/10.1016/j.carbon.2013.07.110> (2013).
- 21 Jian, X. *et al.* Facile Synthesis of Fe₃O₄/GCs Composites and Their Enhanced Microwave Absorption Properties. *ACS Applied Materials & Interfaces* **8**, 6101–6109, doi:10.1021/acsami.6b00388 (2016).
- 22 Li, Y. *et al.* Fe@C nanocapsules with substitutional sulfur heteroatoms in graphitic shells for improving microwave absorption at gigahertz frequencies. *Carbon* **126**, 372–381, doi:<https://doi.org/10.1016/j.carbon.2017.10.040> (2018).
- 23 Song, C. *et al.* Three-dimensional reduced graphene oxide foam modified with ZnO nanowires for enhanced microwave absorption properties. *Carbon* **116**, 50–58, doi:<https://doi.org/10.1016/j.carbon.2017.01.077> (2017).
- 24 Zhao, B. *et al.* Liquid-Metal-Assisted Programmed Galvanic Engineering of Core-shell Nanohybrids for Microwave Absorption. *Advanced Functional Materials* **n/a**, 2302172, doi:<https://doi.org/10.1002/adfm.202302172> (2023).
- 25 Han, X. *et al.* Laser-Induced Graphene from Wood Impregnated with Metal Salts and Use in Electrocatalysis. *ACS Applied Nano Materials* **1**, 5053–5061, doi:10.1021/acsanm.8b01163 (2018).
- 26 Rao, A. M., Eklund, P. C., Bandow, S., Thess, A. & Smalley, R. E. Evidence for charge transfer in doped carbon nanotube bundles from Raman scattering. *Nature* **388**, 257–259, doi:10.1038/40827 (1997).
- 27 Zhou, G. *et al.* Oxygen Bridges between NiO Nanosheets and Graphene for Improvement of Lithium Storage. *ACS Nano* **6**, 3214–3223, doi:10.1021/nn300098m (2012).
- 28 Lei, L. *et al.* Carbon hollow fiber membranes for a molecular sieve with precise-cutoff ultramicropores for superior hydrogen separation. *Nature Communications* **12**, 268, doi:10.1038/s41467-020-20628-9 (2021).
- 29 Chen, J., Wu, X., Tan, Q. & Chen, Y. Designed synthesis of ultrafine NiO nanocrystals bonded on a three dimensional graphene framework for high-

capacity lithium-ion batteries. *New Journal of Chemistry* **42**, 9901-9910,
doi:10.1039/C8NJ01330F (2018).

REVIEWERS' COMMENTS

Reviewer #1 (Remarks to the Author):

Authors have done much effort to expand the technique details and their understanding. The issues raised by the referee are carefully addressed, thus the revision becomes acceptable now. This research focusing on a novel engineering strategy through versatile laser-assisted method, should be further emphasized and highlighted.

Reviewer #2 (Remarks to the Author):

The comments are well addressed.

Reviewer #3 (Remarks to the Author):

The revision has addressed my previous concerns. I would like to recommend the publication of this manuscript in the current form.